# Diminishing Exploration: A Minimalist Approach to Piecewise Stationary Multi-Armed Bandits

## Abstract

The piecewise-stationary bandit problem is an important variant of the multi-armed bandit problem that further considers abrupt changes in the reward distributions. The main theme of the problem is the trade-off between exploration for detecting environment changes and exploitation of traditional bandit algorithms. While this problem has been extensively investigated, existing works either assume knowledge about the number of change points $M$ or require extremely high computational complexity. In this work, we revisit the piecewise-stationary bandit problem from a minimalist perspective. We propose a novel and generic exploration mechanism, called diminishing exploration, which eliminates the need for knowledge about $M$ and can be used in conjunction with an existing change detection-based algorithm to achieve near-optimal regret scaling. Simulation results show that despite being oblivious of $M$, equipping existing algorithms with proposed diminishing exploration generally achieves better empirical regret than the traditional uniform exploration.

## 1 Introduction

The multi-armed bandit (MAB) problem, a classic formulation of online decision making, involves a decision-maker facing a set of arms with unknown reward distributions, and the challenge is to determine a learning strategy that maximizes the cumulative reward. MAB encapsulates the fundamental trade-off between exploiting the current known best arm for immediate reward and exploring other arms for potentially discovering better ones. Given the prevalence of such exploration-exploitation dilemma in practice, MAB has served as the abstraction of various real-world sequential decision making problems, such as recommender systems (Lu et al., 2010), communication networks (Gupta et al., 2019; Hashemi et al., 2018), and clinical trials (Aziz et al., 2021). In the MAB literature, there are two popular frameworks, namely stochastic bandits and adversarial bandits (Lattimore & Szepesvári, 2020): (i) In a standard stochastic bandit model (Lai et al., 1985; Auer et al., 2002a), the rewards of each arm are drawn independently from its underlying reward distribution, which is assumed stationary (i.e., remains fixed throughout the learning process). (ii) In an adversarial model, the reward distribution of each arm is determined by an adversary and could change abruptly after each time step. To extend and unify the above two frameworks, the *piecewise-stationary bandit* (Yu & Mannor, 2009) incorporates *change points* into the bandit model, where the reward distribution of each arm could vary abruptly and arbitrarily at each (unknown) change point and remain fixed between two successive change points. Accordingly, the piecewise-stationary bandit framework serves as a more realistic setting for a broad class of applications which are neither fully stationary nor fully adversarial, such as recommender systems (Xu et al., 2020) and dynamic pricing systems (Yu & Mannor, 2009).

Piecewise-stationary bandit has been extensively studied from various perspectives, including the passive methods, e.g., forgetting via discounting (Kocsis & Szepesvári, 2006) or a sliding window (Garivier & Moulines, 2011), and the active methods that leverage change-point detectors, e.g., (Liu et al., 2018; Cao et al., 2019). A more comprehensive survey is deferred to Section 1.1. Despite the rich literature, the existing approaches suffer from the following issues: (i) *Required knowledge of the number of change points*: To adapt exploration to the change frequency, most of the existing works require some tuning based on the knowledge about the number of change points (denoted by

Table 1: A summary of the regret bounds of various algorithms (R.B.: Regret Bound, S.K.: Segment Knowledge, (DE): Diminishing exploration version, (DE)$^+$: Diminishing exploration extension version, P: Passive method, A: Active method, E: Elimination approach, M: Multiple instances). The notation $S$ is the total number of times the optimal arm switches to another.

| ALG | TYPE | S.K. FREE | COMPLEXITY | R.B. $\tilde{\mathcal{O}}(\cdot)$ | REFERENCE |
|---|---|---|---|---|---|
| D-UCB | P | ✗ | $\mathcal{O}(KT)$ | $\sqrt{MT}$ | (KOCSIS & SZEPESVÁRI, 2006) |
| SW-UCB | P | ✗ | $\mathcal{O}(KT)$ | $\sqrt{MT}$ | (GARIVIER & MOULINES, 2011) |
| D-TS | P | ✗ | $\mathcal{O}(KT)$ | $\sqrt{MT}$ | (QI ET AL., 2023) |
| ADSWITCH | E | ✓ | $\mathcal{O}(KT^4)$ | $\sqrt{MT}$ | (AUER ET AL., 2019) |
| META | E | ✓ | $\mathcal{O}(KT^2)$ | $\sqrt{ST}$ | (SUK & KPOTUFE, 2023) |
| ARMSWITCH | E | ✓ | $\mathcal{O}(K^2T^2)$ | $\sqrt{ST}$ | (ABBASI-YADKORI ET AL., 2023) |
| MASTER | M | ✓ | $\mathcal{O}(KT)$ | $\min\left\{\sqrt{MT}, \Delta^{1/3}T^{2/3} + \sqrt{T}\right\}$ | (WEI & LUO, 2021) |
| CUSUM-UCB | A | ✗ | $\mathcal{O}(KT^2)$ | $\sqrt{MT}$ | (LIU ET AL., 2018) |
| M-UCB | A | ✗ | $\mathcal{O}(KT)$ | $\sqrt{MT}$ | (CAO ET AL., 2019) |
| GLR-KLUCB | A | ✓ | $\mathcal{O}(KT^2)$ | $\sqrt{MT}$ | (BESSON ET AL., 2022) |
| **M-UCB (DE)** | A | ✓ | $\mathcal{O}(KT)$ | $\sqrt{MT}$ | OURS |
| GLR-UCB (DE) | A | ✓ | $\mathcal{O}(KT^2)$ | $\sqrt{MT}$ | OURS |
| **M-UCB (DE)$^+$** | A | ✓ | $\mathcal{O}(KT)$ | $\sqrt{ST}$ | OURS |
| GLR-UCB (DE)$^+$ | A | ✓ | $\mathcal{O}(KT^2)$ | $\sqrt{ST}$ | OURS |

$M$ throughout this paper), which could be rather difficult to obtain or estimate in practice. (ii) *High computational or algorithmic complexity*: AdSwitch (Auer et al., 2019) and its enhanced versions (Suk & Kpotufe, 2022; Abbasi-Yadkori et al., 2023) have been proposed to achieve nearly optimal regret without knowing the number of changes by adopting an elimination approach. However, these approaches are computationally costly as they either relies on a large number of calls of detection mechanism (Auer et al., 2019). On the other hand, MASTER (Wei & Luo, 2021) serves as a generic black-box approach utilizes multiple hierarchical instances to tackle the issue of unknown number of changes and achieve optimal regret guarantees. This additional algorithmic complexity can lead to high overhead and incur high regret, especially for a small number of segments. Moreover, Gerogiannis et al. (2024) emphasizes that MASTER only becomes effective under very large time horizons—a luxury that is often unattainable in practical scenarios. These motivate the need for a minimalist design for piecewise-stationary MAB.

In this paper, we answer the above question through the lens of *diminishing exploration*. Our key insight is that one could achieve a proper trade-off between the detection delay and the regret incurred by exploration if the amount of exploration is configured to decrease with the *elapsed time since the latest detection*, even without the knowledge of $M$.

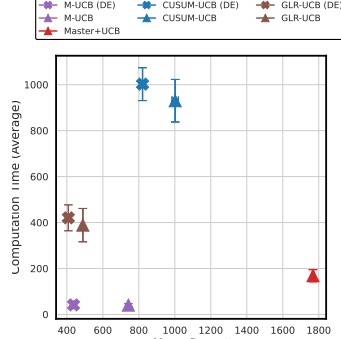

Figure 1: Regret and computation times.

Specifically, the main contribution of this work is to address the piecewise stationary multi-armed bandit problem from a minimalist perspective, reflected in several key aspects:

**Conceptual Simplicity of the Algorithm**: The proposed algorithm is conceptually very simple, which only involves equipping an active method with a novel diminishing exploration mechanism. This allows it to be flexibly used alongside *any* active method, contributing to a clear design logic for the DE+CD+Base algorithm.

**Minimal Knowledge Requirement**: The algorithm operates with minimal knowledge of the environment, requiring no prior information about the number of change points $M$, and still can achieve a nearly-optimal regret bound of $\tilde{\mathcal{O}}(\sqrt{MKT})$ due to its ability to automatically adapt to the environ-

ment. This balance of minimal assumption and strong performance highlights the algorithm's ability to provide excellent regret results numerically.

**Low Complexity**: The algorithm maintains one of the lowest possible complexities, designed without adding extra computational burden. The exploration mechanism is efficiently scheduled, requiring only the determination of when to initiate each exploration phase.

To support our statements, we provide evidence in Figure 1, where the mean regret and computation time across different algorithms are compared. It is shown that the proposed diminishing exploration together with M-UCB can achieve the best performance in terms of both regret and complexity. Compared to MASTER, our algorithm achieves significantly lower regret and substantially reduced computational complexity within a non-asymptotic time horizon—a regime in which MASTER typically underperforms, as analyzed in Gerogiannis et al. (2024). This is an initial observation, and we will discuss it in more detail in Section 6.

This paper is summarized as follows: (i) We revisit piecewise-stationary bandit problems without the knowledge of the number of changes through the lens of diminishing exploration, which is parameter-free, computationally efficient, and compatible with various change detection methods and bandit algorithms. (ii) We provide a general form that allows any change detector to be combined with diminishing exploration and formally show that M-UCB and GLR-UCB equipped with the proposed diminishing exploration scheme under a properly chosen scheduler enjoy $\tilde{\mathcal{O}}(\sqrt{MKT})$ regret bound. Therefore, the proposed algorithm is nearly optimal in terms of dynamic regret without knowing $M$. (iii) Through extensive simulations, we corroborate the regret performance of the proposed algorithm and show that it outperforms the existing benchmark methods in empirical regret.

A summary of the algorithms, which will be reviewed in what follows, and their performance, along with the one proposed in this work, is provided in Table 1. In this table, we highlight the performance of the proposed M-UCB (DE) as evidence of our claim of being an minimalist approach. As shown in the table, it is a nearly optimal algorithm with low computational complexity. In addition, it requires the least knowledge about the environment, offering versatility and ease of extension to M-UCB (DE)$^+$. Last but not least, the performance of GLR-UCB (DE) and its extension GLR-UCB (DE)$^+$ are also provided as an example to demonstrate that the proposed DE can be used in conjunction with active methods, other than M-UCB.

## 1.1 RELATED WORK

**Piecewise-Stationary Bandits With Knowledge of Number of Changes**. The existing bandit algorithms for the piecewise-stationary setting could be largely divided into two categories: (i) *Passive methods*: The forgetting mechanism is one widely adopted technique to tackle piecewise stationarity without explicitly detecting the change points. For example, Discounted UCB (Kocsis & Szepesvári, 2006) and Sliding-Window UCB (Garivier & Moulines, 2011) are two important variants of UCB-type algorithms with respective forgetting mechanism, and they both have been shown to achieve $\mathcal{O}(K\sqrt{MT}\log T)$ dynamic regret. Moreover, (Raj & Kalyani, 2017) propose Discounted Thompson Sampling (DTS), which adapts the discounting technique to the Bayesian setup and enjoys good empirical performance despite the lack of any theoretical guarantee. Subsequently, (Qi et al., 2023) provides valuable insights into DTS with theoretical guarantees, demonstrating that the DTS method can achieve $\mathcal{O}(K\sqrt{MT}\log^2 T)$ regret. By adapting the methods originally designed for adversarial bandits, RExp (Besbes et al., 2014) offers another passive strategy by augmenting the classic Exp3 algorithm (Auer et al., 2002b) with restarts and achieve $\mathcal{O}((K\log KV_T)^{1/3}T^{2/3})$ regret, where $V_T$ denotes the total variation budget up to $T$. (ii) *Active methods*: MAB algorithms augmented with a change-point detector have been explored quite extensively. One example is the Windowed-Mean Shift Detection (WMD) algorithm (Yu & Mannor, 2009), which offers a generic framework of combining change detectors and standard MAB methods. That said, WMD is designed specifically for the setting with additional side information about the rewards of the unplayed arms and hence is not applicable to the standard piecewise-stationary setting. Allesiardo & Féraud (2015) propose Exp3.R, which augments Exp3 with a change detector that resets Exp3 and thereby achieves $\mathcal{O}(NK\sqrt{T\log T})$ with $N$ denoting the number of changes of the best arm ($N = M$ in the worst case). More recently, Liu et al. (2018) propose change-detection based UCB (CD-UCB), which combines UCB method with off-the-shelf change detectors, such as the classic cumulative sum (CUSUM) procedure and Page-Hinkley test. Cao et al. (2019) propose M-UCB, which augments UCB with a simple change detector that is activated when the sample count of an arm reaches a

window size $w$, comparing the summations of the two halves of the samples within the window. Both CD-UCB and M-UCB could achieve the $\mathcal{O}(\sqrt{KMT \log T})$ regret bound. On the other hand, similar ideas have also been studied from a Bayesian perspective (Mellor & Shapiro, 2013; Alami et al., 2017). However, all the methods described above rely on the assumption that $M$ is known. Moreover, piecewise-stationary bandit has also been studied in the constrained setting (Mukherjee, 2022) and the contextual bandit setting (Chen et al., 2019; Zhao et al., 2020).

**Piecewise-Stationary Bandits Without Knowledge of Number of Changes**. Among the existing works, AdSwitch (Auer et al., 2019) and GLR-klUCB (Besson et al., 2022) are the most relevant to ours as they also obviate the need for the knowledge of $M$. Specifically, AdSwitch achieves $\mathcal{O}(\sqrt{KMT \log T})$ regret via an elimination approach, but at the expense of a high computational complexity incurred by a more sophisticated detection scheme. On the other hand, GLR-klUCB employs the Generalized Likelihood Ratio (GLR) test on the samples to detect changes, offering a more efficient detection framework but could achieve an order-optimal regret bound only in easy problem instances. In contrast, our diminishing exploration is meant to acheive optimal regret without the knowledge of $M$ nor strong assumptions on the segment length.

Recent studies on enhancing the practical effectiveness of change detection-based algorithms have considered addressing significant changes without having a complete restart for every detected change. When the reward distributions evolve while the optimal arm remains stable, a full restart is deemed too conservative. For instance, (Manegueu et al., 2021) proposed a change point algorithm based on empirical gaps between arms. (Suk & Kpotufe, 2022) expanded on this by quantifying significant shifts at each step, avoiding reliance on non-stationarity knowledge. They employ a sophisticated method to regularly re-explore suboptimal arms, ensuring optimal guarantees for both piecewise-stationary and variation budget assumptions. (Abbasi-Yadkori et al., 2023) also achieved comparable guarantees in scenarios with abrupt changes, albeit with slightly diminished results. (Wei & Luo, 2021) takes a multi-scale approach and maintains multiple competing instances of the base algorithm, which achieves a good theoretical guarantee.

## 2 PROBLEM FORMULATION

**Piecewise-Stationary Bandit Environment.** A piecewise-stationary bandit can be described using the tuple $\left(\mathcal{K}, \mathcal{T}, \{f_{k,t}\}_{k \in \mathcal{K}, t \in \mathcal{T}}\right)$, where $\mathcal{K}$ represents a set of $K$ arms, $\mathcal{T}$ represents a set of $T$ time steps, and $f_{k,t}$ represents the reward distribution of arm $k \in \mathcal{K}$ in time $t \in \mathcal{T}$. Denote by $X_{k,t}$ the reward provided by the environment at the $t$-th time step if the learner selects the $k$-th arm. This reward is drawn from $f_{k,t}$ independently of the rewards obtained in other time steps $t' \in \mathcal{T}$. At time step $t \in \mathcal{T}$, the learner selects $A_t$, one of the $K$ arms as the action at this time step, and sees a reward $X_{A_t,t}$.

Unlike the stochastic bandit environment, the piecewise-stationary bandit environment has several unknown change points, at which the reward distribution will change. Let us define $M$ as the total number of segments in this piecewise-stationary bandit environment. Mathematically, $M$ can be represented as $M := 1 + \sum_{t=1}^{T-1} \mathbf{1}_{\{f_{k,t} \neq f_{k,t+1} \text{ for any } k \in \mathcal{K}\}}$, where the indicator function $\mathbf{1}_{\{f_{k,t} \neq f_{k,t+1} \text{ for any } k \in \mathcal{K}\}}$ represents the occurrence of a change. We denote the time of the $i$-th change point as $\nu_i$, for all $i \in \{1, \cdots, M-1\}$ and let $\nu_0 = 0$ and $\nu_M = T$. Moreover, we define $s_i := \nu_i - \nu_{i-1}$ as the segment length of each segment $i$. Within the $i$-th segment, for each $k \in \mathcal{K}$, the reward distribution $f_{k,t}$ are the same for all $t \in [\nu_{i-1} + 1, \nu_i]$; therefore from this point onward, we slightly abuse the notation to simply use $f_k^{(i)}$ and $\mu_k^{(i)}$ to denote the reward distribution and the corresponding expected value, respectively, for arm $k$ in the $i$-th segment.

**Regret Minimization.** Similar to Liu et al. (2018); Cao et al. (2019), we adopt the *dynamic regret* as the performance metric:

$$\mathcal{R}(T) := \sum_{t=1}^{T} \max_{k \in \mathcal{K}} \mathbb{E}[X_{k,t}] - \mathbb{E}\left[\sum_{t=1}^{T} X_{A_t,t}\right]. \tag{1}$$

In the context of piecewise-stationary bandit, our objective, like in other bandit problems, is to minimize regret. Bandit algorithms, such as UCB, are known for solving the tension between exploration and exploitation in stationary bandit problems. The main challenge in piecewise-stationary

bandit lies again in the trade-off between exploration and exploitation. To illustrate, in each segment, after sufficiently exploring the environment, a traditional bandit algorithm will (perhaps softly) commit to the current known best arm. This commitment is difficult to break in the presence of unnoticed change. Additional exploration can be invested to identify changes for resetting the algorithm, which inevitably introduces additional regret. This gives rise to a new exploration and exploitation trade-off that investigates how much additional exploration should be conducted. Specifically, the more additional exploration, the quicker the changes are detected, leading to a quicker reset of the algorithm. The goal of this work is to solve this tension by proposing a novel exploration mechanism that can strike an optimal balance between the regret incurred by additional exploration and that due from detection delay.

## 3 THE PROPOSED FRAMEWORK: DIMINISHING EXPLORATION

In this section, we provide the proposed algorithm in detail. We note that in most of the active algorithms for piecewise-stationary bandit problems, such as Yu & Mannor (2009); Liu et al. (2018); Cao et al. (2019), there is a (periodic or stochastic) uniform exploration scheme which spends a constant fraction of time on exploration for detecting potential changes, together with a traditional algorithm that is capable of attaining near-optimal tradeoff for the traditional bandit problem. We referred to this type of algorithms as a change detection (CD)-based bandit algorithm. Our proposed algorithm is a novel and generic exploration technique, called diminishing exploration, which can be used in conjunction with a CD-based bandit algorithm.

### 3.1 DIMINISHING EXPLORATION

The motivation of the proposed diminishing exploration lies in the following two observations about the uniform exploration: 1) The uniform exploration scheme spends a constant fraction of time on exploration, which results in a regret proportional to the configured exploration rate. 2) To determine an exploration rate that achieves the

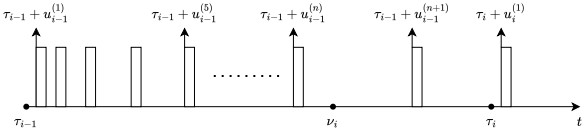

Figure 2: Diminishing exploration.

optimal regret scaling, the information about the total number of segments (or change points) is required. To address the above issues, we propose the diminishing exploration scheme as follows. Let us define $\tau_i$ as the $i$-th time when the algorithm alarms a change. In the proposed method, a uniform exploration round starts at $u_{i-1}^{(j)}$ for $j \in \{1, 2, \ldots\}$ with $u_{i-1}^{(1)} = \tau_{i-1} + 1$. i.e., the learner chooses to pull the arm $1, 2, \ldots, K$ at time $u_{i-1}^{(j)}, u_{i-1}^{(j)} + 1, \ldots, u_{i-1}^{(j)} + K - 1$, respectively. The process restarts whenever a new change $\tau_i$ is detected.

We aim to balance the regret resulting from exploration and that associated with the performance of change detection by dynamically adjusting the exploration rate *within a segment*. Let $u_{i-1}^{(j)}$ be the start time of the $j$-th uniform exploration session between two consecutive alarms $\tau_i$ and $\tau_{i-1}$. In our approach, these sessions are designed in such a way that $u_{i-1}^{(j+1)} - u_{i-1}^{(j)}$ is greater than $u_{i-1}^{(j)} - u_{i-1}^{(j-1)}$. This means that the inter-session time within the same time segment increases with $j$, which in turn results in reduction in the exploration rate. Specifically, for the $i$-th segment, we choose $u_i^{(1)} = \left\lceil (\alpha - K/4\alpha)^2 \right\rceil$ and $u_i^{(j)} = \left\lceil u_i^{(j-1)} + \frac{K}{\alpha}\sqrt{u_i^{(j-1)}} + \frac{K^2}{4\alpha^2} \right\rceil$, $\forall 1 \le i \le M, j \ge 2$, without the knowledge of $M$ and the parameter $\alpha$ will be chosen later. Clearly, we have $u_i^{(j-1)} + K < u_i^{(j)}$ for every $j \ge 2$; thus, these exploration phases will not overlap. Moreover, it is obvious that the duration between two exploration phases $u_i^{(j)} - u_i^{(j-1)} = \mathcal{O}(\sqrt{u_i^{(j-1)}})$ increases with time as Figure 2; hence, the exploration rate decreases. Thus, we term this mechanism *diminishing exploration*.

### 3.2 INTEGRATING OFF-THE-SHELF CHANGE DETECTORS WITH DIMINISHING EXPLORATION

The proposed algorithm is given in Algorithm 1, which adopts the proposed diminishing exploration (lines 3-8) and executes traditional UCB (lines 9-12) otherwise. Moreover, the algorithm enters the change detection subroutine (lines 17-19) whenever accumulating sufficient observations for an arm (line 18).

Before concluding this section, we reemphasize that although we selected as an example to employ the change detection algorithm in Algorithm 3 (Cao et al., 2019) and 4 (Besson et al., 2022) in Algorithm 1 together with the proposed diminishing exploration, the diminishing exploration technique can, in fact, be used in conjunction with any CD-based algorithm.

## 4 REGRET ANALYSIS

In this section, to show the effectiveness of the proposed diminishing exploration, we define two sets of events to capture the behavior of the change point detection algorithm: The false alarm events are defined as $F_i := \{\tau_i < \nu_i\}$, $\forall 1 \leq i \leq M-1$, and $F_0 := \{\tau_0 = 0\}$; the event that the detection delay of the $i$-th change is smaller than $h_i$ is defined as $D_i := \{\tau_i \leq \nu_i + h_i\}$, $\forall 1 \leq i \leq M-2$, where the choice of $h_i$ depends on the underlying CD algorithm. We also define $D_0 := \{\tau_0 = 0\}$, and $D_{M-1} := \{\tau_{M-1} \leq T\}$. In our regret analysis, we also define the following two quantities $\Delta_k^{(i)} := \max_{\tilde{k} \in \mathcal{K}} \left\{ \mu_{\tilde{k}}^{(i)} \right\} - \mu_k^{(i)}$, $\forall 1 \leq i \leq M$, $k \in \mathcal{K}$, and $\delta_k^{(i)} := \left| \mu_k^{(i+1)} - \mu_k^{(i)} \right|$, $\forall 1 \leq i \leq M-1$, $k \in \mathcal{K}$. Furthermore, let $\delta^{(i)} := \max_{k \in \mathcal{K}} \delta_k^{(i)}$.

**Theorem 4.1.** *The Algorithm 1 can be combined with a CD algorithm, which achieves the expected regret upper bound as follows:*

---

**Algorithm 1** CD-UCB with diminishing exploration

**Require:** Positive integer $T, K$ and parameter $\alpha$
1: Initialize $\tau \leftarrow 0$, $u \leftarrow \left\lceil (\alpha - K/4\alpha)^2 \right\rceil$ and $n_k \leftarrow 0 \; \forall k \in \mathcal{K}$;
2: **for** $t = 1, 2, \ldots, T$ **do**
3:   **if** $u \leq t - \tau < u + K$ **then**
4:     $A_t \leftarrow (t - \tau) - u + 1$
5:   **else**
6:     **if** $t - \tau = u + K$ **then**
7:       $u \leftarrow \left\lceil u + \frac{K}{\alpha}\sqrt{u} + \frac{K^2}{4\alpha^2} \right\rceil$
8:     **end if**
9:     **for** $k = 1, \ldots, K$ **do**
10:       $\text{UCB}_k \leftarrow \frac{1}{n_k} \sum_{n=1}^{n_k} Z_{k,n} + \sqrt{\frac{2 \log(t - \tau)}{n_k}}$
11:     **end for**
12:     $A_t \leftarrow \arg\max_{k \in \mathcal{K}} \text{UCB}_k$
13:   **end if**
14:   Play arm $A_t$ and receive the reward $X_{A_t, t}$.
15:   $n_{A_t} \leftarrow n_{A_t} + 1; Z_{A_t, n_{A_t}} \leftarrow X_{A_t, t}$
16:   **if** CD = True **then**
17:     $\tau \leftarrow t$, $u \leftarrow 1$ and $n_k \leftarrow 0 \; \forall k \in \mathcal{K}$
18:   **end if**
19: **end for**

---

$$\mathbb{E}\left[R(1, T)\right] \leq \underbrace{\sum_{i=1}^{M} \tilde{C}_i}_{(a)} + \underbrace{2\alpha\sqrt{MT}}_{(b)} + \underbrace{\sum_{i=1}^{M-1} \mathbb{E}\left[\tau_i - \nu_i \big| D_i \overline{F}_i D_{i-1} \overline{F}_{i-1}\right]}_{(c)}$$

$$+ \underbrace{T \sum_{i=1}^{M} \mathbb{P}\left(F_i \big| \overline{F}_{i-1} D_{i-1}\right) + T \sum_{i=1}^{M-1} \mathbb{P}\left(\overline{D}_i \big| \overline{F}_i \overline{F}_{i-1} D_{i-1}\right)}_{(d)}, \quad (2)$$

*where* $\tilde{C}_i = 8 \sum_{\Delta_k^{(i)} > 0} \frac{\log T}{\Delta_k^{(i)}} + \left(\frac{5}{2} + \frac{\pi^2}{3} + K\right) \sum_{k=1}^{K} \Delta_k^{(i)}$.

To elaborate, let us look into each term in equation 2. As shown in Lemma C.3, term (a) bounds the regret of the UCB algorithm in each stationary segment, given that the CD algorithm successfully detected the previous change. Term (b) bounds the regret incurred from the diminishing exploration, as shown in Lemma C.1. The other two terms bound the regrets incurred in the phase of change detection, whose quantities would depend on the underlying CD algorithm. Specifically, term (c) corresponds to the regret associated with the detection delay while term (d) addresses the regret from unsuccessful detection and false alarm. For a more detailed proof, see Appendix C.

### 4.1 Integration with change detectors

In this section, we will integrate change detectors from M-UCB and GLR-UCB into the framework of diminishing exploration. Through theoretical analysis, we will demonstrate that diminishing exploration can be extended to other change detectors and achieve a nearly optimal regret bound. All the proofs are deferred to Appendix C.

**Integration with the change detector of M-UCB.** In M-UCB, change detectors are triggered when the sample count of an arm reaches a window size $w$. The change detector divides the samples in the window into two halves and compares the difference between the two halves' summations. If the result exceeds a threshold $b$, an alarm is raised. We define $h_0 := 0$ and choose $h_i = \left\lceil w\left(K/2\alpha + 1\right)\sqrt{s_i + 1} + w^2/4\left(K/2\alpha + 1\right)^2 \right\rceil$ and make the following assumption:

**Assumption 4.2.** The algorithm knows a lower bound $\delta > 0$ such that $\delta \leq \min_i \max_{k \in \mathcal{K}} \delta_k^{(i)}$.

Note that Assumption 4.2 is Assumption 1(b) of Cao et al. (2019), which is required for the M-UCB detector to determine good $w$ and $b$ in regret analysis. It is worth noting that almost all schemes that actively detect changes share similar assumptions; however, different algorithms may impose distinct sets of assumptions. This assumption is mild since $\delta$ may be statistically derived from historical information. Furthermore, even if the lower bound does not hold, and we occasionally encounter changes with expected reward gaps smaller than the assumed $\delta$, such changes may be perceived as too minor to result in significant regret. In Section 6, this fact will be verified through simulation.

With this assumption, we analyze the regret of Algorithm 3 with $w$ and $b$ given by

$$w = \left(4/\delta^2\right) \cdot \left[\sqrt{\log\left(2KT^2\right)} + \sqrt{\log\left(2T\right)}\right]^2, \tag{3}$$

$$b = \left[w\log\left(2KT^2/2\right)\right]^{1/2}. \tag{4}$$

**Assumption 4.3.** $s_i = \Omega\left(\left(\log KT + \sqrt{K\log KT}\right)\sqrt{s_{i-1}}\right)$.

In particular, if $s_i = \Theta\left(\left(\log KT + \sqrt{K\log KT}\right)^{2(1+\epsilon)}\right)$ for every $i$, Assumption 4.3 holds. This assumption essentially posits that the changes are not overly dense, a condition that holds in many practical scenarios. Simple math would then show that given $D_{i-1}$ is true, with this assumption and the proposed diminishing exploration, each arm will have at least $w/2$ observations before and after a change point. Again, we note that similar assumptions are imposed in other algorithms that actively detect changes, while different algorithms may impose different assumptions. This assumption is necessary with our proof technique, which requires every change to be successfully detected with high probability. In our simulations in Section 6, we will demonstrate that when this assumption is violated, all the considered active methods will experience similar performance degradation due to the overly dense changes and our algorithm is not particularly vulnerable. In fact, in our simulation results in Section 6, we show that our algorithms significantly outperform existing active methods, even when this assumption is violated.

**Corollary 4.4** (Regret bound of M-UCB). *Algorithm 1 integrated with Algorithm 3 with the parameters in (3) and (4) achieves the expected regret upper bound of $\mathcal{O}(\sqrt{KMT\log T})$.*

**Integration with the change detector of GLR-UCB.** In GLR-UCB, the Generalized Likelihood Ratio (GLR) test is employed on the samples to detect changes. i.e., an alarm is raised whenever the GLR statistic exceeds a threshold $\beta$ given by

$$\beta = 2\mathcal{J}\left(\frac{\log\left(3T^2\right)}{2}\right) + 6\log\left(1 + \log T\right), \tag{5}$$

where the function $\mathcal{J}$ is defined in Appendix C.1.2. Following (Besson et al., 2022), we define $h_0 := 0$ and choose $h_i = (\alpha, \epsilon) := \left\lceil 2\left(\frac{4}{(\delta^{(i)})^2}\beta + 2\right)\left(\frac{K}{2\alpha} + 1\right)\sqrt{s_i + 1} + \left(\frac{4}{(\delta^{(i)})^2}\beta + 2\right)^2\left(\frac{K}{2\alpha} + 1\right)^2 \right\rceil$ and make the following:

**Assumption 4.5.** $\nu_i - \nu_{i-1} \geq 2\max\left\{h_i, h_{i-1}\right\}$ for all $i \in \{1, \ldots, M\}$.

Under this assumption, we prove the following:

**Corollary 4.6** (Regret bound of GLR-UCB). *Algorithm 1 integrated with Algorithm 4 with $\beta$ in (5) achieves the expected regret upper bound as $\mathcal{O}(\sqrt{KMT\log T})$.*

**Discussion.** In some literature, such as Liu et al. (2018) and Cao et al. (2019), the approach involves finding an exploration rate for uniform exploration that balances regret induced by exploration and detection delay, assuming knowledge of $M$. In GLR-UCB Besson et al. (2022), the exploration rate increases with the number of change detection alarms generated by the algorithm. Compared to other exploration mechanisms, the distinctive feature of *diminishing exploration* is its use of a variable exploration rate within the same segment. The greatest advantage of this approach lies in the fact that it does not require knowledge of $M$. Moreover, its complexity remains low, and it can be readily applied to other active methods.

## 5 EXTENSION TO DETECTION OF OPTIMAL ARM CHANGES

Let us define $S$ as the number of *super-segments*, each of which is the time period between two consecutive changes of the optimal arm. Mathematically, $S$ can be represented as $S := 1 + \sum_{t=1}^{T-1} \mathbf{1}_{\left\{\arg\max_{k\in\mathcal{K}} \mu_{k,t} \neq \arg\max_{k\in\mathcal{K}} \mu_{k,t+1}\right\}}$, where the indicator function $\mathbf{1}_{\left\{\arg\max_{k\in\mathcal{K}} \mu_{k,t} \neq \arg\max_{k\in\mathcal{K}} \mu_{k,t+1}\right\}}$ represents the occurrence of the optimal arm changing to another one. We denote the time of the $r$-th occurrence of the optimal arm changing to another one as $\nu_r^*$, for all $r \in \{1, \cdots, S-1\}$ and let $\nu_0^* = 0$ and $\nu_S^* = T$. Moreover, we define $s_r^* := \nu_r^* - \nu_{r-1}^*$ as the super segment length of each segment $r$, which is the duration for which the optimal arm remains the same. The last, we define $\tau_r^*$ as the $r$-th time when the algorithm alarms an optimal arm changing to another one. We have provided Figure 5 to visually clarify the differences from Section 4.

Similar to Section 4, we also define two sets of events to capture the behavior of the change point detection algorithm in the version where we only focus on the change of the optimal arm: The false alarm events are defined as $F_r^* := \{\tau_r^* < \nu_r^*\}$, $\forall\, 1 \leq r \leq S-1$, and $F_0^* := \{\tau_0^* = 0\}$; the event that the detection delay of the $r$-th change is smaller than $h_r^*$ is defined as $D_r^* := \{\tau_r^* \leq \nu_r^* + h_r^*\}$, $\forall\, 1 \leq r \leq S-2$, where the choice of $h_r^*$ depends on the CD algorithm. We also define $D_0^* := \{\tau_0^* = 0\}$, and $D_{S-1}^* := \{\tau_{S-1}^* \leq T\}$. In our regret analysis, we also define the following quantities $\Delta_{k,t} := \max_{\tilde{k}\in\mathcal{K}} \left\{\mu_{\tilde{k},t}\right\} - \mu_{k,t}$, $\forall\, 1 \leq t \leq T$, $k \in \mathcal{K}$, and assume $\Delta_{\min} := \min_{k\in\mathcal{K}} \min_{1\leq t\leq T} \Delta_{k,t}$ is known.

*Remark* 5.1. In this section, the definition of a false alarm differs from that in Section 4. Here, a false alarm occurs when we incorrectly claim a change in the optimal arm. Consequently, even if an arm's distribution changes but the optimal arm remains unchanged, any such alarm would still be considered a false alarm.

---

**Algorithm 2** Skipping Mechanism

---

**Require:** Two positive integer $n_{\text{skip},k}$ and $n_{\text{skip},k^*}$, $n_{\text{skip},k}$ observations $X_1, \ldots, X_{n_{\text{skip},k}}$ and $n_{\text{skip},k^*}$ observations $Y_1, \ldots, Y_{n_{\text{skip},k^*}}$.

1: **if** $\quad \sum_{\ell=1}^{n_{\text{skip},k}} X_\ell / n_{\text{skip},k} \quad <$ $\sum_{\ell=1}^{n_{\text{skip},k^*}} Y_\ell / n_{\text{skip},k^*} + \eta$ **then**
2:     Return True
3: **else**
4:     Return False
5: **end if**

---

### 5.1 DIMINISHING EXPLORATION WITH A SKIPPING MECHANISM

Here, we introduce a skipping mechanism (see Algorithm 2 for the high-level concept and Algorithm 5 in Appendix B for details) to ignore unnecessary alarms. The algorithm takes two sets of observations of size $n_{\text{skip},k}$ and $n_{\text{skip},k^*}$, respectively, as inputs and checks whether the sample average of the second set is larger than that of the first set by a margin of $\eta$. In Appendix B, we present the complete algorithm, where the two sets are samples of our algorithm before and after an alarm of change, respectively. If Algorithm 2 returns true, then this alarm is skipped; otherwise, it declares that the optimal arm has changed and resets the algorithm. Note that having a negative $\eta$ would reduce miss detection but may also increase false reset. On the other hand, having a positive $\eta$ would encourage skipping, reducing false reset but leading to higher miss detection. Besides, the optimal $\eta$ also depends heavily on the underlying CD algorithm, as some CD algorithms cause higher false alarm rates than others. In our numerical (in Appendix F) and analytic results (Theorem C.13 in Appendix C.2), we set $n_{\text{skip},k} = \mathcal{O}(\log T)$ and $n_{\text{skip},k^*} = \mathcal{O}(\log T)$ and demonstrate the effectiveness of the proposed skipping mechanism with $\eta = 0$. When applying the skipping mechanism, minor adjustments may be necessary depending on the specific change detector employed. For instance, for some change detectors, whenever a change in an arm's distribution is identified, regardless of whether it is skipped, the data within the change detector's buffer is cleared. The detailed algorithm is provided in Appendix B.

### 5.2 INTEGRATION WITH CHANGE DETECTORS

In this section, we will integrate change detectors from M-UCB and GLR-klUCB into the extension of diminishing exploration similar to Section 4.

**Integration with change detectors of M-UCB.** We choose the parameter $w$ as

$$w = \left(8/\min\{\delta, \Delta_{min}\}^2\right) \cdot \left[\sqrt{\log\left(2KT^2\right)} + \sqrt{\log\left(2T\right)}\right]^2. \tag{6}$$

The selection of the remaining parameters is the same as in Section 4.

**Corollary 5.2** (Regret bound of M-UCB). *Combining Algorithm 1 and 3 with the parameters in Equation 4, and Equation 6 achieves the expected regret upper bound as $\mathcal{O}(\sqrt{KST \log T})$.*

**Integration with change detectors of GLR-UCB.** The selection of parameters is the same as in Section 4.

**Corollary 5.3** (Regret bound of GLR-UCB). *Combining Algorithm 1 and 4 with $\beta$ function in Equation 5 achieves the expected regret upper bound as $\mathcal{O}(\sqrt{KST \log T})$.*

The proofs for Corollary 5.2 and 5.3 are in Appendix C.2.

## 6 SIMULATION RESULTS

In this section, we assess the effectiveness of the proposed diminishing exploration scheme across various dimensions, encompassing regret scaling in $M$, $K$, and $T$, regrets in synthetic environments, and regrets in a real-world scenario. In addition to evaluating M-UCB (Cao et al., 2019) with our diminishing exploration, we also examine a variant of CUSUM-UCB (Liu et al., 2018) that incorporates diminishing exploration with CUSUM-UCB, further highlighting the efficacy of the proposed exploration method. We will compare our approach with M-UCB, CUSUM-UCB, GLR-UCB, Discounted-UCB, Discounted Thompson Sampling, and MASTER. Unless stated otherwise, we report the average regrets over 100 simulation trials, with a zoomed-in view of the figures provided in G. Detailed configuration is provided in Appendix D.

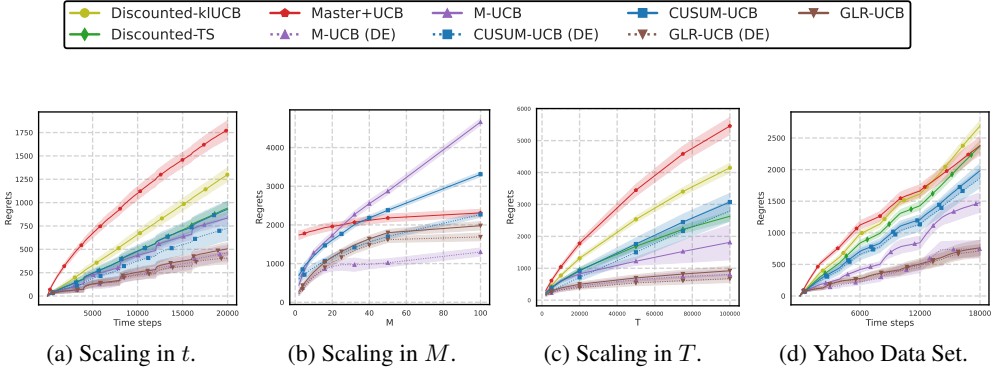

(a) Scaling in $t$.      (b) Scaling in $M$.      (c) Scaling in $T$.      (d) Yahoo Data Set.

Figure 3: Regret in the synthetic environments and under the Yahoo data set.

**Regret in Each Time Step.** In this simulation, we consider a multi-armed bandit problem with $T = 20000$ time steps and $M = 5$. Recall that $\mu_k^{(i)}$ represents the expected value for arm $k$ in the $i$-th segment. Here, we set $\mu_k^{(i)} = 0.2, 0.5, 0.8$ for $i$ with $(i + k) \mod 3 = 2, 0, 1$, respectively. Figure 3a shows that for both CUSUM-UCB and M-UCB, employing diminishing exploration can effectively reduce the additional regret caused by constant exploration. Moreover, M-UCB with the proposed diminishing exploration achieves the lowest regret. In the figure, the change points are clearly evident by observing the breakpoints in each line. The reason for the overall steeper slope of CUSUM-UCB (both with and without diminishing exploration) is due to the heightened sensitivity of the CUSUM detector itself, resulting in more frequent false alarms.

**Regret Scaling in $M$.** Based on the settings outlined above, with the only variation being in the parameter $M$, Figure 3b illustrates the dynamic regrets across various values of $M$. In this experiment, adjustments to the exploration parameter settings are required based on the size of $M$ when using a constant exploration rate. However, this is not the case for the proposed diminishing exploration. The result confirms the earlier-discussed rationale that the proposed diminishing exploration can automatically adapt to the environment, resulting in the best regret performance among the algorithms.

**Regret Scaling in $T$.** In line with the setting described above, with the only variation being the parameter $T$, we present the dynamic regrets across different values of $T$. Observations similar to those made above can again be found in Figure 3c, where the proposed diminishing exploration can effectively reduce the regret.

**Regret and Execution Time.** Here, we compare the computation time and regret across different algorithms for various choices of $M$ and $T$ with other parameters same as above. Specifically, we

conduct experiments under three scenarioso: one where the environment changes rapidly ($M = 50$ and $T = 20000$), one where the environment changes slowly ($M = 5$ and $T = 20000$), and one where the considered time horizon is double ($M = 5$ and $T = 40000$). As shown in Figure 4a to 4c, despite oblivious of $M$, our algorithm almost always achieves the lowest computation time and regret in all the scenarios. Moreover, comparing to Master+UCB, another algorithm not requiring the knowledge of $M$, our algorithm is always significantly better as shown in these figures. Figure 4d plots the ratio of average execution time of Master+UCB to that of M-UCB with our DE for various $T$. It is shown that the growth rate is faster than $0.5 \log T$, and it appears to become even linear in $T$ as $T$ increases.

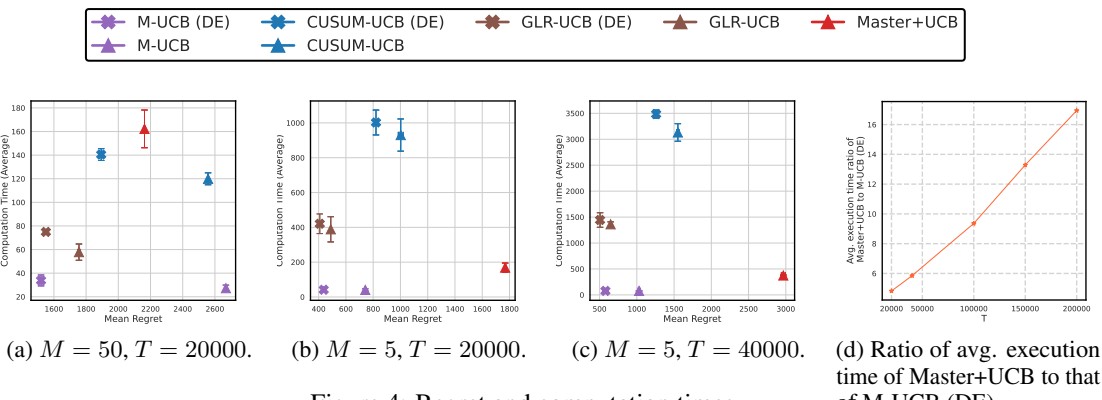

(a) $M = 50, T = 20000$.  (b) $M = 5, T = 20000$.  (c) $M = 5, T = 40000$.  (d) Ratio of avg. execution time of Master+UCB to that of M-UCB (DE).

Figure 4: Regret and computation times.

**Regret in an Environment Built from a Real-World Dataset.** We further utilize the benchmark dataset publicly published by Yahoo! for evaluation. To enhance arm distinguishability in our simulation, we scale up the data by a factor of 10. The number of segments is set to $M = 9$ and the number of arms is set to $K = 6$. Figure 3d shows the evolution of dynamic regret. Again, we see that the diminishing exploration scheme could help M-UCB, GLR-UCB and CUSUM-UCB achieve similar or better regret even without knowing $M$.

**Scenarios when Assumptions are Violated.** In Assumption 4.2, we assumes the knowledge of $\delta$ to select an appropriate $w$. We emphasize that our settings in many of the above simulations actually violate this assumption. Take Figure 3a for example, $w = 200$ is chosen, which corresponds to $\delta \approx 0.6$ when back calculating, which is much larger than the actual $\delta = 0.3$ of this scenario. Assumptions 4.3 and 4.5 provide guarantees that the segment length is sufficiently long. However, in the last data point of our Figure 3b ($M = 100$), these assumptions are clearly violated. In this case, it becomes challenging for the active methods to promptly detect every change point. For algorithms like M-UCB and CUSUM-UCB, which require knowledge of $M$, their awareness of quick changes causes them to invest more effort into change detection, leading to a very high exploration rate. However, this does not always guarantee successful detection, resulting in very high regret. Diminishing exploration, on the other hand, continues to decrease the exploration rate regardless of $M$ when no changes are detected. This allows more resources to be invested for UCB, which might gradually adapt to the environment's changes, leading to a regret that is lower than that of uniform exploration.

Regarding the simulations of AdSwitch, ArmSwitch, and the Meta algorithm, due to the extremely long running time, we have not been able to finish the simulation for $T$ beyond 20000. We alternatively perform comparison with smaller $T$, whose results are presented in Appendix F.

## 7 CONCLUDING REMARKS

In this paper, we revisited the piecewise-stationary bandit problem. A novel diminishing exploration mechanism, called diminishing exploration, was proposed that does not require knowledge about the number of stationary segments. When used in conjunction with the M-UCB and GLR-UCB, the proposed diminishing exploration mechanism was rigorously shown to achieve a near optimal regret. Extensive simulations were also provided to show the effectiveness of the proposed mechanism. Regarding the limitations, since the proposed diminishing exploration is employed together with a CD algorithm, the integrated algorithm usually inherits from the CD algorithm a constraint on the length of each segment in order to guarantee near-optimal regret performance.

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

# A  CHANGE DETECTION ALGORITHM

**Change Detector of M-UCB**
The following algorithm is the change detection algorithm for M-UCB (Cao et al., 2019)

---
**Algorithm 3** Change Detection of M-UCB: $\text{CD}(w, b, Z_1, \ldots, Z_w)$

---
**Require:** An even integer $w$, $w$ observations $Z_1, \ldots, Z_w$, and a prescribed
    threshold $b > 0$
1: **if** $\left| \sum_{\ell=w/2+1}^{w} Z_\ell - \sum_{\ell=1}^{w/2} Z_\ell \right| > b$ **then**
2:    Return True
3: **else**
4:    Return False
5: **end if**

---

In this algorithm, one requires $w$ observations as input and check whether the difference between the sample average of the first half and that of the second half exceeds a prescribed threshold $b$ (line 1).

**Change Detector of GLR-UCB**
The following definition is the change detection algorithm for GLR-UCB (Besson et al., 2022)

**Definition A.1.** The Bernoulli GLR change-point detector with threshold function $\beta(n, \epsilon)$ is

$$\tau_\delta := \inf \left\{ n \in \mathbb{N}^* : \sup_{s \in [1,n]} \left[ s \times \text{kl}(\hat{\mu}_{1:s}, \hat{\mu}_{1:n}) + (n - s) \times \text{kl}(\hat{\mu}_{s+1:n}, \hat{\mu}_{1:n}) \right] \geq \beta(n, \delta) \right\} \quad (7)$$

---
**Algorithm 4** Change Detection of GLR-UCB: $\text{CD}(Y_1, \ldots, Y_{n_k})$

---
**Require:** $Y_1, \ldots, Y_{n_k}$, and a threshold function $\beta(n, \epsilon)$
1: **if** $\sup_{s \in [1,n]} \left[ s \times \text{kl} \left( \sum_{\ell=1}^{s} Y_\ell / s, \sum_{\ell=1}^{n} Y_\ell / n \right) + (n - s) \times \text{kl} \left( \sum_{\ell=s+1}^{n} Y_\ell / (n - s), \sum_{\ell=1}^{n} Y_\ell / n \right) \right] \geq$
   $\beta(n, \delta)$ **then**
2:    Return True
3: **else**
4:    Return False
5: **end if**

---

# B  THE EXTENDED VERSION ALGORITHM

The skipping mechanism in more detail is as follows:

---

**Algorithm 5** Skipping mechanism $(Z, n, n_{\text{skip}}, N_I)$

---

**Require:** An array $Z$ with elements $Z_{k,t}$ for $k \in \mathcal{K}$ and $t \in \{1, \ldots, n_k\}$; two vectors $n$ and $n_{\text{skip}}$ with elements $n_k$ and $n_{\text{skip},k}$ for $k \in \mathcal{K}$; and a positive integer $N_I$.

1: $k^* \leftarrow \arg\max_{k \in \mathcal{K}} n_k$
2: **if** $\exists k \neq k^* \in \mathcal{K}, n_{\text{skip},k} \geq N_I$ and $\sum_{\ell=1}^{n_{\text{skip},k}} Z_{k,n_k - n_{\text{skip},k}+1}/n_{\text{skip},k} > \sum_{\ell=1}^{n_{\text{skip},k^*}} Z_{k,n_{k^*} - n_{\text{skip},k^*}+1}/n_{\text{skip},k^*} + \eta$ **then**
3:     Return False
4: **else**
5:     Return True
6: **end if**

---

As mentioned in Section 5, some adjustments may be required depending on the specific change detector used. Here, we provide the algorithm 6 to illustrate such adjustments.

The characteristics of the M-UCB detector allow for a relatively straightforward integration with the skipping mechanism; therefore, our algorithm can be simplified as algorithm 7.

---

**Algorithm 6** CD-UCB with diminishing exploration and skipping mechanism

---

**Require:** Positive integer $T, K$, and parameter $\alpha, N_I$

1: Initialize $\tau \leftarrow 0$, $u \leftarrow \left\lceil (\alpha - K/4\alpha)^2 \right\rceil$, $n_k, n_{\text{CD},A_t} \leftarrow 0 \ \forall k \in \mathcal{K}$ and skip_mode $\leftarrow$ False;

2: **for** $t = 1, 2, \ldots, T$ **do**

3:     **if** skip_mode = True **then**

4:         **if** $A_{\text{change}} = k^*$ **then**

5:             $A_t \leftarrow t \bmod K$

6:         **else**

7:             **if** $t$ is even **then**

8:                 $A_t \leftarrow k^*$

9:             **else**

10:                $A_t \leftarrow A_{\text{change}}$

11:             **end if**

12:         **end if**

13:         $n_{\text{skip},A_t} \leftarrow n_{\text{skip},A_t} + 1$

14:     **else**

15:         **if** $u \leq t - \tau < u + K$ **then**

16:             $A_t \leftarrow (t - \tau) - u + 1$

17:         **else**

18:             **if** $t - \tau = u + K$ **then**

19:                 $u \leftarrow \left\lceil u + \frac{K}{\alpha}\sqrt{u} + \frac{K^2}{4\alpha^2} \right\rceil$

20:             **end if**

21:             **for** $k = 1, \ldots, K$ **do**

22:                 $\text{UCB}_k \leftarrow \frac{1}{n_k} \sum_{n=1}^{n_k} Z_{k,n} + \sqrt{\frac{2\log(t-\tau)}{n_k}}$

23:             **end for**

24:             $A_t \leftarrow \arg\max_{k \in \mathcal{K}} \text{UCB}_k$

25:         **end if**

26:     **end if**

27:     Play arm $A_t$ and receive the reward $X_{A_t,t}$.

28:     $n_{A_t} \leftarrow n_{A_t} + 1$; $Z_{A_t,n_{A_t}} \leftarrow X_{A_t,t}$

29:     $n_{\text{CD},A_t} \leftarrow n_{\text{CD},A_t} + 1$; $Z_{\text{CD},A_t,n_{\text{CD},A_t}} \leftarrow X_{A_t,t}$

30:     **if** CD = True **then**

31:         **if** skip_mode = False **then**

32:             $k^* \leftarrow \arg\max_{k \in \mathcal{K}} n_k$

33:         **end if**

34:         $A_{\text{change}} \leftarrow A_t$

35:         skip_mode $\leftarrow$ True

36:         $n_{\text{skip},k} \leftarrow 0 \ \forall k \in \mathcal{K}$

37:         $n_{\text{CD},k} \leftarrow 0 \ \forall k \in \mathcal{K}$

38:     **end if**

39:     **if** skip_mode = True **then**

40:         **if** $\text{Skip}(Z, n, n_{\text{skip}}, N_I)$ = False **then**

41:             $\tau \leftarrow t$, $u \leftarrow 1$ and $n_k, n_{\text{skip},k} \leftarrow 0 \ \forall k \in \mathcal{K}$

42:             skip_mode $\leftarrow$ False

43:         **end if**

44:         **if** $(A_{\text{change}} = k^*$ and $\sum_{k \in \mathcal{K}} n_{\text{skip},k} > KN_I)$ or $(A_{\text{change}} \neq k^*$ and $\sum_{k \in \mathcal{K}} n_{\text{skip},k} > 2N_I)$ **then**

45:             $n_{\text{skip},k} \leftarrow 0 \ \forall k \in \mathcal{K}$

46:             skip_mode $\leftarrow$ False

47:         **end if**

48:     **end if**

49: **end for**

---

---

**Algorithm 7** M-UCB with diminishing exploration and skipping mechanism

---

**Require:** Positive integer $T, K$, and parameter $\alpha$

1: Initialize $\tau \leftarrow 0$, $u \leftarrow \left\lceil (\alpha - K/4\alpha)^2 \right\rceil$ and $n_k \leftarrow 0 \ \forall k \in \mathcal{K}$;

2: **for** $t = 1, 2, \ldots, T$ **do**

3:     **if** $u \leq t - \tau < u + K$ **then**

4:         $A_t \leftarrow (t - \tau) - u + 1$

5:     **else**

6:         **if** $t - \tau = u + K$ **then**

7:             $u \leftarrow \left\lceil u + \frac{K}{\alpha}\sqrt{u} + \frac{K^2}{4\alpha^2} \right\rceil$

8:         **end if**

9:         **for** $k = 1, \ldots, K$ **do**

10:             $\text{UCB}_k \leftarrow \frac{1}{n_k}\sum_{n=1}^{n_k} Z_{k,n} + \sqrt{\frac{2\log(t-\tau)}{n_k}}$

11:         **end for**

12:         $A_t \leftarrow \arg\max_{k \in \mathcal{K}} \text{UCB}_k$

13:     **end if**

14:     Play arm $A_t$ and receive the reward $X_{A_t, t}$.

15:     $n_{A_t} \leftarrow n_{A_t} + 1$; $Z_{A_t, n_{A_t}} \leftarrow X_{A_t, t}$

16:     **if** $n_{A_t} \geq w$ **then**

17:         **if** $\text{CD}(w, b, Z_{A_t, n_{A_t} - w + 1}, \ldots, Z_{A_t, n_{A_t}}) = \text{True}$ **then**

18:             **if** $\text{Skip}(Z, n, n, w/2) = \text{False}$ **then**

19:                 $\tau \leftarrow t$, $u \leftarrow 1$ and $n_k, n_{\text{skip},k} \leftarrow 0 \ \forall k \in \mathcal{K}$

20:             **end if**

21:         **end if**

22:     **end if**

23: **end for**

---

# C   PROOF DETAIL

This appendix provides the detailed proofs of the results presented in Section 4 and Section 5. Each proof is carefully elaborated to ensure clarity and rigor.

## C.1   PROOF OF SECTION 4

In this subsection, we present the proofs of Theorem 4.1 in Section 4. In what follows, the first lemma bounds the regret accumulated during the diminishing exploration part of the algorithm. We denote by $R_{\text{DE}}(\tau_{i-1}, \nu_i)$ as the regret caused by the exploration part of the algorithm and by $N_{\text{DE},k}(\tau_{i-1}, \nu_i)$ the number of times that the arm $k$ is selected in the exploration phase from the previous alarm time to the next change point.

**Lemma C.1** (Diminishing exploration regret). *If the mean values of the arms remain the same during the time interval $[\tau_{i-1}, \nu_i)$, then we have*

$$N_{\text{DE},k}(\tau_{i-1}, \nu_i) \leq \frac{2\alpha\sqrt{\nu_i - \tau_{i-1}}}{K} + \frac{3}{2}, \tag{8}$$

*and*

$$\mathbb{E}\left[R_{\text{DE}}(\tau_{i-1}, \nu_i)\right] \leq 2\alpha\sqrt{\nu_i - \tau_{i-1}} + \frac{3}{2}K. \tag{9}$$

*Proof.* Recall that $u_i^{(j)}$ is the beginning of the $j$-th uniform exploration session in the $i$-th segment. In Algorithm 1, the initial time of the first exploration session after each $\tau_i$ is given by:

$$u_i^{(1)} = \left\lceil \left(\alpha - \frac{K}{4\alpha}\right)^2 \right\rceil, \tag{10}$$

and subsequent times follow the recursive equation:

$$u_i^{(j)} = \left\lceil u_i^{(j-1)} + \frac{K}{\alpha}\sqrt{u_i^{(j-1)}} + \frac{K^2}{4\alpha^2} \right\rceil \geq u_i^{(j-1)} + \frac{K}{\alpha}\sqrt{u_i^{(j-1)}} + \frac{K^2}{4\alpha^2}. \tag{11}$$

Based on equation 10 and equation 11, one could easily check that the sequence $u_i^{(n)}$ satisfies that for every natural number $n$,

$$u_i^{(n)} \geq \left( \frac{(2n-3)K}{4\alpha} + \alpha \right)^2. \tag{12}$$

Let $u_i^{(m)}$ be the last exploration start time in time interval $[\tau_{i-1}, \nu_i)$. Then, we have

$$\mathbb{E}\left[R_{\mathrm{DE}}\left(\nu_i - \tau_{i-1}\right)\right] \leq mK. \tag{13}$$

Additionally, we have:

$$\nu_i - \tau_{i-1} \geq u_i^{(m)} \geq \left( \frac{(2m-3)K}{4\alpha} + \alpha \right)^2 \geq \left( \frac{2\mathbb{E}\left[R_{DE}\left(\nu_i - \tau_{i-1}\right)\right] - 3K}{4\alpha} + \alpha \right)^2. \tag{14}$$

Finally, based on the equation 13 and equation 14, we can conclude that:

$$\mathbb{E}\left[R_{\mathrm{DE}}\left(\nu_i - \tau_{i-1}\right)\right] \leq 2\alpha\sqrt{\nu_i - \tau_{i-1}} - 2\alpha^2 + \frac{3}{2}K \leq 2\alpha\sqrt{\nu_i - \tau_{i-1}} + \frac{3}{2}K, \tag{15}$$

and

$$N_{\mathrm{DE},k}\left(\nu_i - \tau_{i-1}\right) \leq \frac{2\alpha\sqrt{\nu_i - \tau_{i-1}}}{K} + \frac{3}{2}. \tag{16}$$

$\square$

In the following lemma, we aim to explore how long it takes for a given arm to reach a certain number of samples through diminishing exploration.

**Lemma C.2** (Samples-time steps transform). *When each arm has accumulated $n$ samples, the required time is as follows: If the counting of the $n$ samples begins immediately after a reset, the required time is given by*

$$T_{reset} \leq \left( \alpha + \frac{(2n-3)K}{4\alpha} + n \right)^2 + K. \tag{17}$$

*However, if there is a delay of $t_d$ time steps after the reset before the counting begins, the required time is given by*

$$T_{t_d} \leq 2n\left( \frac{K}{2\alpha} + 1 \right)\sqrt{t_d + 1} + n^2\left( \frac{K}{2\alpha} + 1 \right)^2. \tag{18}$$

*Proof.* We can derive the following from Equation 11 in the proof of Lemma C.1:

$$u^{(j)} \leq u^{(j-1)} + \frac{k}{\alpha}\sqrt{u^{(j-1)}} + \frac{k^2}{4\alpha^2} + 1 \leq \left( \sqrt{u^{(j-1)}} + \frac{k}{2\alpha} \right)^2 + 1 \leq \left( \sqrt{u^{(j-1)}} + \frac{k}{2\alpha} + 1 \right)^2. \tag{19}$$

First, let us consider the case where the counting of $n$ samples begins immediately after the reset. From Equation 10, we can derive the following:

$$u^{(1)} \leq \left( \alpha - \frac{K}{4\alpha} + 1 \right)^2. \tag{20}$$

Using Equation 19, we can further derive:

$$u^{(2)} \leq \left( \alpha - \frac{K}{4\alpha} + 1 + \frac{K}{2\alpha} + 1 \right)^2. \tag{21}$$

Finally, we obtain:

$$u^{(n)} \leq \left( \alpha - \frac{K}{4\alpha} + 1 + (n-1)\left(\frac{K}{2\alpha} + 1\right) \right)^2. \tag{22}$$

Here, $u^{(n)}$ represents the starting time of the $n$-th exploration block. The total time required to guarantee that all $K$ arms have been sampled $n$ times is therefore:

$$T_{\text{reset}} = u^{(n)} + K \leq \left( \alpha + \frac{(2n-3)K}{4\alpha} + n \right)^2 + K. \tag{23}$$

Next, we consider how long it takes for each arm to be sampled $n$ times after $t_d$ time steps following the reset. We first assume that, prior to $t_d$, each arm has already been sampled $x$ times. For simplicity, we assume an ideal case where the exploration block starts exactly at time $t_d + 1$. Therefore, we have:

$$u^{(x+1)} = t_d + 1. \tag{24}$$

Since, in reality, the exploration block start time may not exactly coincide with $t_d + 1$, we account for the possibility that it could begin at a later time by considering the next exploration block's start time. This allows us to bound the non-ideal case.

Following the same approach as in the first part of the proof, we eventually obtain:

$$u^{(x+n+1)} \leq \left( \sqrt{t_d + 1} + n\left(\frac{K}{2\alpha} + 1\right) \right)^2. \tag{25}$$

Finally, we derive that the total time required for each arm to be sampled $n$ times after $t_d$ time steps is:

$$T_{t_d} = u^{(x+n+1)} - u^{(x+1)} \leq 2n\left(\frac{K}{2\alpha} + 1\right)\sqrt{t_d + 1} + n^2\left(\frac{K}{2\alpha} + 1\right)^2. \tag{26}$$

$$\square$$

Define $R(r, s) := \sum_{t=r}^{s} \max_{k \in \mathcal{K}} \mathbb{E}[X_{k,t}] - X_{A_t,t}$ be the regret accumulated during $r$ and $s$. In the next lemma, we provide an upper bound on the regret accumulated from the $(i-1)$-th alarm time to the end of $(i-1)$-th segment, given that the previous change was successfully detected.

**Lemma C.3** (Regret bound with stationary bandit). *Consider a stationary bandit interval with $\nu_{i-1} < \tau_{i-1} < \nu_i$. Condition on the successful detection events $\overline{F}_{i-1}$ and $D_{i-1}$, the expected regret accumulated during $(\tau_{i-1}, \nu_i)$ can be bounded by*

$$\mathbb{E}\left[R(\tau_{i-1}, \nu_i) | \overline{F}_{i-1} D_{i-1}\right] \leq \tilde{C} + 2\alpha\sqrt{s_i} + T \cdot \mathbb{P}\left(F_i | \overline{F}_{i-1} D_{i-1}\right), \tag{27}$$

*where $\tilde{C}$ is as described in Theorem 4.1.*

*Proof.* For every $i$, we have

$$\mathbb{E}\left[R(\tau_{i-1}, \nu_i) | \overline{F}_{i-1} D_{i-1}\right] = \mathbb{E}\left[R(\tau_{i-1}, \nu_i) | F_i \overline{F}_{i-1} D_{i-1}\right] \mathbb{P}\left(F_i | \overline{F}_{i-1} D_{i-1}\right) \tag{28a}$$

$$+ \mathbb{E}\left[R(\tau_{i-1}, \nu_i) | \overline{F}_i \overline{F}_{i-1} D_{i-1}\right] \mathbb{P}\left(\overline{F}_i | \overline{F}_{i-1} D_{i-1}\right) \tag{28b}$$

$$\leq T \cdot \mathbb{P}\left(F_i | \overline{F}_{i-1} D_{i-1}\right) + \mathbb{E}\left[R(\tau_{i-1}, \nu_i) | \overline{F}_i \overline{F}_{i-1} D_{i-1}\right]. \tag{28c}$$

Now, define $N_k(t_1, t_2) := \sum_{t=t_1}^{t_2} \mathbf{1}_{\{A_t = k\}}$ to be the number of times that arm $k$ is selected by Algorithm 1 from $t_1$ to $t_2$. Note that

$$\mathbb{E}\left[R(\tau_{i-1}, \nu_i) | \overline{F}_i \overline{F}_{i-1} D_{i-1}\right] = \sum_{\Delta_k^{(i)} > 0} \Delta_k^{(i)} \cdot \mathbb{E}\left[N_k(\tau_{i-1}, \nu_i) | \overline{F}_i \overline{F}_{i-1} D_{i-1}\right]. \tag{29}$$

To bound the second term of equation 28c, we further bound $N_k\left(\tau_{i-1}, \nu_i\right)$ as follows,

$$N_k\left(\tau_{i-1}, \nu_i\right) = \sum_{t=\tau_{i-1}+1}^{\nu_i} \mathbf{1}_{\{A_t=k, \tau_i>\nu_i, N_k(\tau_{i-1}, \nu_i)<l\}} + \sum_{t=\tau_{i-1}+1}^{\nu_i} \mathbf{1}_{\{A_t=k, \tau_i>\nu_i, N_k(\tau_{i-1}, \nu_i)\geq l\}}$$
(30a)

$$\leq l + N_{DE,k}\left(\nu_i - \tau_{i-1}\right) + \sum_{t=\tau_{i-1}+1}^{\nu_i} \mathbf{1}_{\left\{k=\arg\max_{k\in\mathcal{K}} \mathrm{UCB}_{k\in\mathcal{K}}, \tau_i>\nu_i, N_k(\tau_{i-1}, \nu_i)\geq l\right\}}$$
(30b)

$$\leq l + \frac{2\alpha\sqrt{\nu_i - \tau_{i-1}}}{K} + \frac{3}{2} + \sum_{t=\tau_{i-1}+1}^{\nu_i} \mathbf{1}_{\left\{k=\arg\max_{k\in\mathcal{K}} \mathrm{UCB}_{k\in\mathcal{K}}, \tau_i>\nu_i, N_k(\tau_{i-1}, \nu_i)\geq l\right\}}$$
(30c)

$$\leq l + \frac{2\alpha\sqrt{s_i}}{K} + \frac{3}{2} + \sum_{t=\tau_{i-1}+1}^{\nu_i} \mathbf{1}_{\left\{k=\arg\max_{k\in\mathcal{K}} \mathrm{UCB}_{k\in\mathcal{K}}, \tau_i>\nu_i, N_k(\tau_{i-1}, \nu_i)\geq l\right\}},$$
(30d)

Equation 30c follows from Lemma C.1. Setting $l = \left\lceil 8\log T/\left(\Delta_k^{(i)}\right)^2 \right\rceil$, and following the same steps as in the proof of Theorem 1 of Auer et al. (2002a), we arrive at

$$\mathbb{E}\left[N_k\left(\tau_{i-1}, \nu_i\right)\middle|\overline{F}_i\overline{F}_{i-1}D_{i-1}\right] \leq \frac{2\alpha\sqrt{s_i}}{K} + \frac{8\log T}{\left(\Delta_k^{(i)}\right)^2} + \frac{5}{2} + \frac{\pi^2}{3} + K.$$
(31)

Putting everything together completes the proof. □

Theorem 4.1 can then be proved by recursively applying Lemma C.3.

**Theorem 4.1** The Algorithm 1 can be combined with a CD algorithm, which achieves the expected regret upper bound as follows:

$$\mathbb{E}\left[R\left(1, T\right)\right] \leq \sum_{i=1}^M \tilde{C}_i + 2\alpha\sqrt{MT} + \sum_{i=1}^{M-1} \mathbb{E}\left[\tau_i - \nu_i\middle|D_i\overline{F}_i D_{i-1}\overline{F}_{i-1}\right]$$

$$+ T\sum_{i=1}^M \mathbb{P}\left(F_i\middle|\overline{F}_{i-1}D_{i-1}\right) + T\sum_{i=1}^{M-1} \mathbb{P}\left(\overline{D}_i\middle|\overline{F}_i\overline{F}_{i-1}D_{i-1}\right), \quad (32)$$

where $\tilde{C}_i = 8\sum_{\Delta_k^{(i)}>0} \frac{\log T}{\Delta_k^{(i)}} + \left(\frac{5}{2} + \frac{\pi^2}{3} + K\right)\sum_{k=1}^K \Delta_k^{(i)}$.

*Proof.* Recall that $R\left(r, s\right) = \sum_{t=r}^s \max_{k\in\mathcal{K}} \mathbb{E}\left[X_{k,t}\right] - X_{A_t,t}$, then $\mathcal{R}\left(T\right) = \mathbb{E}\left[R\left(1, T\right)\right]$. We have

$$\mathcal{R}\left(T\right) = \mathbb{E}\left[R\left(1, T\right)\right]$$
(33a)

$$= \mathbb{E}\left[R\left(1, T\right)\middle|\overline{F}_0 D_0\right]$$
(33b)

$$\leq \mathbb{E}\left[R\left(1, \nu_1\right)\middle|\overline{F}_1\overline{F}_0 D_0\right] + \mathbb{E}\left[R\left(\nu_1, T\right)\middle|\overline{F}_1\overline{F}_0 D_0\right] + T\cdot\mathbb{P}\left(F_1\middle|\overline{F}_0 D_0\right)$$
(33c)

$$\leq \tilde{C}_1 + 2\alpha\sqrt{\left(\nu_1 - \nu_0\right)} + \mathbb{E}\left[R\left(\nu_1, T\right)\middle|\overline{F}_1\overline{F}_0 D_0\right] + T\cdot\mathbb{P}\left(F_1\middle|\overline{F}_0 D_0\right),$$
(33d)

where equation 33b holds because $\tau_0 = 0$, equation 33c is due to the law of total expectation and some trivial bounds, and equation 33d follows from Lemmas C.3. The third term in equation 33d is then further bounded as follows:

$$\mathbb{E}\left[R\left(\nu_1,T\right)\middle|\overline{F}_1\overline{F}_0D_0\right] \leq \mathbb{E}\left[R\left(\nu_1,T\right)\middle|D_1\overline{F}_1\overline{F}_0D_0\right] + T\cdot\left(1-\mathbb{P}\left(D_1\middle|\overline{F}_1\overline{F}_0D_0\right)\right) \tag{34a}$$

$$\leq \mathbb{E}\left[R\left(\nu_1,T\right)\middle|D_1\overline{F}_1\overline{F}_0D_0\right] + T\cdot\mathbb{P}\left(\overline{D}_1\middle|\overline{F}_1\overline{F}_0D_0\right) \tag{34b}$$

$$= \mathbb{E}\left[R\left(\tau_1,T\right)\middle|D_1\overline{F}_1\overline{F}_0D_0\right] + \mathbb{E}\left[R\left(\nu_1,\tau_1\right)\middle|D_1\overline{F}_1\overline{F}_0D_0\right] + T\cdot\mathbb{P}\left(\overline{D}_1\middle|\overline{F}_1\overline{F}_0D_0\right) \tag{34c}$$

$$\leq \mathbb{E}\left[R\left(\tau_1,T\right)\middle|\overline{F}_1D_1\right] + \mathbb{E}\left[\tau_1-\nu_1\middle|\overline{F}_1D_1\overline{F}_0D_0\right] + T\cdot\mathbb{P}\left(\overline{D}_1\middle|\overline{F}_1\overline{F}_0D_0\right) \tag{34d}$$

$$\leq \mathbb{E}\left[R\left(\tau_1,T\right)\middle|\overline{F}_1D_1\right] + \mathbb{E}\left[\tau_1-\nu_1\middle|\overline{F}_1D_1\right] + T\cdot\mathbb{P}\left(\overline{D}_1\middle|\overline{F}_1\overline{F}_0D_0\right), \tag{34e}$$

where equation 34a applies the law of total expectation and some trivial bounds. From here, we can set up the following recursion:

$$\mathbb{E}\left[R\left(1,T\right)\right] = \mathbb{E}\left[R\left(1,T\right)\middle|\overline{F}_0D_0\right] \tag{35a}$$

$$\leq \mathbb{E}\left[R\left(\tau_1,T\right)\middle|\overline{F}_1D_1\right] + \tilde{C}_1 + 2\alpha\sqrt{s_1-1} + \mathbb{E}\left[\tau_1-\nu_1\middle|\overline{F}_1D_1\right] \tag{35b}$$

$$+ T\cdot\mathbb{P}\left(F_1\middle|\overline{F}_0D_0\right) + T\cdot\mathbb{P}\left(\overline{D}_1\middle|\overline{F}_1\overline{F}_0D_0\right) \tag{35c}$$

$$\leq \mathbb{E}\left[R\left(\tau_2,T\right)\middle|\overline{F}_2D_2\right] + \sum_{i=1}^{2}\tilde{C}_i + 2\alpha\sum_{i=1}^{2}\sqrt{s_i-1} \tag{35d}$$

$$+ \sum_{i=1}^{2}\mathbb{E}\left[\tau_i-\nu_i\middle|\overline{F}_{i-1}D_{i-1}\right] + T\sum_{i=1}^{2}\mathbb{P}\left(F_i\middle|\overline{F}_{i-1}D_{i-1}\right) + T\sum_{i=1}^{2}\mathbb{P}\left(\overline{D}_i\middle|\overline{F}_i\overline{F}_{i-1}D_{i-1}\right) \tag{35e}$$

$$\vdots$$

$$\leq \sum_{i=1}^{M}\tilde{C}_i + 2\alpha\sum_{i=1}^{M}\sqrt{s_i} + \sum_{i=1}^{M-1}\mathbb{E}\left[\tau_i-\nu_i\middle|\overline{F}_{i-1}D_{i-1}\right] \tag{35f}$$

$$+ T\sum_{i=1}^{M}\mathbb{P}\left(F_i\middle|\overline{F}_{i-1}D_{i-1}\right) + T\sum_{i=1}^{M-1}\mathbb{P}\left(\overline{D}_i\middle|\overline{F}_i\overline{F}_{i-1}D_{i-1}\right) \tag{35g}$$

$$\leq \sum_{i=1}^{M}\tilde{C}_i + 2\alpha\sqrt{MT} + \sum_{i=1}^{M-1}\mathbb{E}\left[\tau_i-\nu_i\middle|\overline{F}_{i-1}D_{i-1}\right] \tag{35h}$$

$$+ T\sum_{i=1}^{M}\mathbb{P}\left(F_i\middle|\overline{F}_{i-1}D_{i-1}\right) + T\sum_{i=1}^{M-1}\mathbb{P}\left(\overline{D}_i\middle|\overline{F}_i\overline{F}_{i-1}D_{i-1}\right) \tag{35i}$$

where equation 35h follows from the Cauchy–Schwarz inequality

$$\left(\sum_{i=1}^{M}\sqrt{s_i}\right)^2 \leq \left(\sum_{i=1}^{M}s_i\right)\left(\sum_{i=1}^{M}1\right) = M\sum_{i=1}^{M}s_i = MT, \tag{36a}$$

$\square$

### C.1.1 PROOF OF INTEGRATION WITH CHANGE DETECTORS OF M-UCB

With the general regret bound in Theorem 4.1, a regret bound of the M-UCB with the proposed diminishing exploration can be obtained by bounding $\mathbb{P}\left(F_i\middle|\overline{F}_{i-1}D_{i-1}\right)$, $\mathbb{P}\left(D_i\middle|\overline{F}_i\overline{F}_{i-1}D_{i-1}\right)$, and $\mathbb{E}\left[\tau_i-\nu_i\middle|\overline{F}_iD_i\overline{F}_{i-1}D_{i-1}\right]$.

First, in Lemma C.4, we show that the probability of false alarm is very small; thereby, its contribution to the regret is negligible.

**Lemma C.4** (Probability of false alarm). *Under Algorithm 1 with parameter in equation 3, and equation 4, we have*

$$\mathbb{P}\left(F_i\middle|\overline{F}_{i-1}D_{i-1}\right) \leq wK\left(1-\left(1-\exp\left(-2b^2/w\right)\right)^{\lfloor T/w\rfloor}\right) \leq \frac{1}{T}. \tag{37}$$

*Proof.* Suppose that at time $t$, we have gathered $w$ samples of arm $k \in \mathcal{K}$, namely $Y_{k,1}, Y_{k,2}, \ldots, Y_{k,w}$, for change detection in line 17 of Algorithm 1, and we define

$$S_{k,t} = \sum_{\ell=w/2+1}^{w} Y_{k,\ell} - \sum_{\ell=1}^{w/2} Y_{k,\ell}. \tag{38}$$

Note that $S_{k,t} = 0$ when there is insufficient (less than $w$) samples to trigger the change detection algorithm. By definition, we have

$$\tau_{k,i} = \inf\{t \geq \tau_{i-1} + w : |S_{k,t}| > b\}. \tag{39}$$

Given that the events $D_{i-1}$ and $\bar{F}_{i-1}$ hold, we define $\tau_{k,i}$ as the first detection time of the $k$-th arm after $\nu_i$. Clearly, $\tau_i = \min_{k \in \mathcal{K}} \{\tau_{k,i}\}$ as Algorithm 1 would reset every time a change is detected. Using the union bound, we have

$$\mathbb{P}\left(F_i \middle| \overline{F}_{i-1} D_{i-1}\right) = \mathbb{P}\left(\max_{k \in \mathcal{K}} \sum_{t=\tau_{i-1}+1}^{\nu_i} \mathbf{1}_{\{A_t=k\}} \geq w, F_i \middle| \overline{F}_{i-1} D_{i-1}\right) \tag{40a}$$

$$+ \mathbb{P}\left(\max_{k \in \mathcal{K}} \sum_{t=\tau_{i-1}+1}^{\nu_i} \mathbf{1}_{\{A_t=k\}} < w, F_i \middle| \overline{F}_{i-1} D_{i-1}\right) \tag{40b}$$

$$= \mathbb{P}\left(F_i \middle| \overline{F}_{i-1} D_{i-1}, \max_{k \in \mathcal{K}} \sum_{t=\tau_{i-1}+1}^{\nu_i} \mathbf{1}_{\{A_t=k\}} \geq w\right) \tag{40c}$$

$$\cdot \mathbb{P}\left(\max_{k \in \mathcal{K}} \sum_{t=\tau_{i-1}+1}^{\nu_i} \mathbf{1}_{\{A_t=k\}} \geq w \middle| \overline{F}_{i-1} D_{i-1}\right) \tag{40d}$$

$$\leq \mathbb{P}\left(F_i \middle| \overline{F}_{i-1} D_{i-1}, \max_{k \in \mathcal{K}} \sum_{t=\tau_{i-1}+1}^{\nu_i} \mathbf{1}_{\{A_t=k\}} \geq w\right) \tag{40e}$$

$$\leq \sum_{k=1}^{K} \mathbb{P}\left(\tau_{k,i} \leq \nu_i \middle| \overline{F}_{i-1} D_{i-1}, \max_{k' \in \mathcal{K}} \sum_{t=\tau_{i-1}+1}^{\nu_i} \mathbf{1}_{\{A_t=k'\}} \geq w\right) \tag{40f}$$

$$\leq \sum_{k=1}^{K} \mathbb{P}\left(\tau_{k,i} \leq \nu_i \middle| \overline{F}_{i-1} D_{i-1}, \sum_{t=\tau_{i-1}+1}^{\nu_i} \mathbf{1}_{\{A_t=k\}} \geq w\right), \tag{40g}$$

where the term in equation 40b is clearly equal to $0$ as there will be no false alarm if we do not even have sufficiently many observations to trigger the alarm as suggested by Algorithm 3. Equation 40c and equation 40d hold by the definition of conditional probability, equation 40e is due to the fact that the term in equation 40d is at most one, and equation 40f follows from the union bound. In equation 40g, if $k \neq k'$, we cannot guarantee that $\sum_{t=\tau_{i-1}+1}^{\nu_i} \mathbf{1}_{\{A_t=k'\}} \geq w$. Hence, some $k$ might cause the probability in the equation 40f to be zeros.

For any $0 \leq j \leq w - 1$, define the stopping time

$$\tau_{k,i}^{(j)} := \inf\{t = \tau_{i-1} + j + nw, n \in \mathbb{Z}^+ : |S_{k,t}| > b\}. \tag{41}$$

Clearly, $\tau_{k,i} = \min\{\tau_{k,i}^{(0)}, \ldots, \tau_{k,i}^{(w-1)}\}$. Let us define, for any $0 \leq j \leq w - 1$,

$$\xi_{k,i}^{(j)} = \frac{\left(\tau_{k,i}^{(j)} - j - \tau_{i-1}\right)}{w}. \tag{42}$$

Note that condition on the events $D_{i-1}$ and $\bar{F}_{i-1}$, $\xi_{k,i}^{(j)}$ is a geometric random variable with parameter $p := \mathbb{P}(|S_{k,t}| > b)$, because when fixing $j$, there is no overlap between the samples in the current

window and the next.

$$\mathbb{P}\left(\tau_{k,i}^{(j)} = \tau_{i-1} + nw + j \middle| \overline{F}_{i-1}D_{i-1}, \sum_{t=\tau_{i-1}+1}^{\nu_i} \mathbf{1}_{\{A_t=k\}} \geq w\right)$$

$$= \mathbb{P}\left(\xi_{k,i} = n \middle| \overline{F}_{i-1}D_{i-1}, \sum_{t=\tau_{i-1}+1}^{\nu_i} \mathbf{1}_{\{A_t=k\}} \geq w\right) = p(1-p)^{n-1}. \quad (43)$$

Here, the inclusion of subsequent events as conditions should not impact the results, as when entering the change detection algorithm, those events have already occurred. Moreover, by union bound, we have that for any $k \in \mathcal{K}$,

$$\mathbb{P}\left(\tau_{k,i} \leq \nu_i \middle| \overline{F}_{i-1}D_{i-1}, \sum_{t=\tau_{i-1}+1}^{\nu_i} \mathbf{1}_{\{A_t=k\}} \geq w\right) \leq w\left(1 - (1-p)^{\lfloor(\nu_i-\tau_{i-1})/w\rfloor}\right) \quad (44a)$$

$$\leq w\left(1 - (1-p)^{\lfloor T/w\rfloor}\right). \quad (44b)$$

We further use the McDiarmid's inequality and the union bound to show that

$$p = \mathbb{P}\left(|S_{k,t}| > b\right) = \mathbb{P}\left(S_{k,t} > b\right) + \mathbb{P}\left(S_{k,t} < -b\right) \quad (45a)$$

$$\leq 2 \cdot \exp\left(-\frac{2b^2}{w}\right). \quad (45b)$$

Using the result in equation 44b and equation 45b into equation 40g,

$$\mathbb{P}\left(F_i \middle| \overline{F}_{i-1}D_{i-1}\right) \leq \sum_{k=1}^{K} w\left(1 - \left(1 - 2\exp\left(-\frac{2b^2}{w}\right)\right)^{\lfloor T/w\rfloor}\right) \quad (46a)$$

$$= wK\left(1 - \left(1 - 2\exp\left(-\frac{2b^2}{w}\right)\right)^{\lfloor T/w\rfloor}\right). \quad (46b)$$

Moreover, applying $(1-x)^a > 1 - ax$ for any $a > 1$ and $0 < x < 1$ and plugging the choice of $b = \sqrt{w\log\left(2KT^2\right)/2}$ as in equation 4 shows the second inequality. $\qquad\square$

Lemma C.2 ensures that, with high probability, the detection delay is confined within a tolerable interval. That is, each arm is sampled $w/2$ times, and using equation 18 from lemma C.2, we select $h_i$ as

$$h_i = \left\lceil w\left(\frac{K}{2\alpha} + 1\right)\sqrt{s_i+1} + \frac{w^2}{4}\left(\frac{K}{2\alpha}+1\right)^2\right\rceil. \quad (47)$$

**Lemma C.5** (Probability of successful detection)**.** *Consider a piecewise-stationary bandit environment. For any $\mu^{(i)}, \mu^{(i+1)} \in [0,1]^K$ with parameters chosen in equation 3 and equation 4 and*

$$h_i = \left\lceil w\left(\frac{K}{2\alpha} + 1\right)\sqrt{s_i+1} + \frac{w^2}{4}\left(\frac{K}{2\alpha}+1\right)^2\right\rceil, \quad (48)$$

*for some $k \in \mathcal{K}, i \geq 1$ and $c > 0$, under the Algorithm 1, we have*

$$\mathbb{P}\left(D_i \middle| \overline{F}_i\overline{F}_{i-1}D_{i-1}\right) \geq 1 - \frac{1}{T}. \quad (49)$$

*Proof.*

$$\mathbb{P}\left(D_i\big|\overline{F}_i\overline{F}_{i-1}D_{i-1}\right) = \mathbb{P}\left(\tau_i \leq \nu_i + h_i\big|\overline{F}_i\overline{F}_{i-1}D_{i-1}\right) \tag{50a}$$

$$\geq \max_{t\in\{\nu_i+1,\ldots,\nu_i+h_i\}} \mathbb{P}\left(S_{\tilde{k},t} > b\big|\overline{F}_i\overline{F}_{i-1}D_{i-1}\right) \tag{50b}$$

$$\geq \max_{j\in\{0,\ldots,w/2\}} \left(1 - 2\exp\left(-\frac{(j\left|\delta_{\tilde{k}}^{(i)}\right| - b)^2}{w}\right)\right) \tag{50c}$$

$$= 1 - 2\exp\left(-\frac{(w|\delta_{\tilde{k}}^{(i)}|/2 - b)^2}{w}\right) \tag{50d}$$

$$\geq 1 - 2\exp\left(-\frac{wc^2}{4}\right). \tag{50e}$$

where $S_{\tilde{k},t}$ is defined in equation 38, equation 50c follows from the McDiarmid's inequality, and equation 50d is due to the fact that the maximum value is attained when $j = w/2$. Last, equation 50e is true for any choice of $w, b$ and $c$ such that $\delta_{\tilde{k}}^{(i)} \geq 2b/w + c$ holds. We thus set $w$ and $b$ as in equation 3 and equation 4, respectively, and choose $c = 2\sqrt{\log(2T)/w}$, which leads to $\mathbb{P}\left(D_i\big|\overline{F}_i\overline{F}_{i-1}D_{i-1}\right) \geq 1 - 1/T$. $\qquad\square$

Lemma C.6 further bounds the expected detection delay in the situation where the change detection algorithm successfully detects the change within the desired interval.

**Lemma C.6** (Expected detection delay). *Consider a piecewise-stationary bandit environment. For any $\mu^{(i)}, \mu^{(i+1)} \in [0,1]^K$ with parameters chosen in equation 3 and equation 4 and*

$$h_i = \left\lceil w\left(\frac{K}{2\alpha} + 1\right)\sqrt{s_i + 1} + \frac{w^2}{4}\left(\frac{K}{2\alpha} + 1\right)^2 \right\rceil, \tag{51}$$

*for some $K \in \mathcal{K}, i \geq 1$ and $c > 0$, under the Algorithm 1, we have*

$$\mathbb{E}\left[\tau_i - \nu_i\big|\overline{F}_iD_i\overline{F}_{i-1}D_{i-1}\right] \leq h_i. \tag{52}$$

*Proof.* For any $1 \leq i \leq M$, we have

$$\mathbb{E}\left[\tau_i - \nu_i\big|\overline{F}_iD_iG_i\overline{F}_{i-1}D_{i-1}\right] = \sum_{j=1}^{h_i}\mathbb{P}\left(\tau_i \geq \nu_i + j\big|\overline{F}_iD_iG_i\overline{F}_{i-1}D_{i-1}\right) \leq h_i. \tag{53a}$$

$\qquad\square$

Plugging the bounds in Lemmas C.4, C.5 and C.6 into Theorem 4.1 shows the following regret bound in Corollary 4.4.

**Corollary 4.4** Combining Algorithm 1 and 3 with the parameters in Equation 3, and Equation 4 achieves the expected regret upper bound as follows:

$$\mathbb{E}[R(1,T)] \leq \underbrace{\sum_{i=1}^{M}\tilde{C}_i}_{(a)} + \underbrace{2\alpha\sqrt{MT}}_{(b)} + \underbrace{w\left(\frac{K}{2\alpha} + 1\right)\sqrt{M(T+M)}}_{(c)}$$

$$+ \underbrace{\frac{w^2M}{4}\left(\frac{K}{2\alpha} + 1\right)^2}_{(c)} + \underbrace{2M}_{(d)}, \tag{54}$$

where $\tilde{C}_i = 8\sum_{\Delta_k^{(i)}>0}\frac{\log T}{\Delta_k^{(i)}} + \left(\frac{5}{2} + \frac{\pi^2}{3} + K\right)\sum_{k=1}^{K}\Delta_k^{(i)}$. By setting $\alpha = c\sqrt{K\log(KT)}$ for some constant $c$, the expected regret is upper-bounded by $\mathcal{O}(\sqrt{KMT\log T})$.

### C.1.2 PROOF OF INTEGRATION WITH CHANGE DETECTORS OF GLR-UCB

First, we introduce the function $\mathcal{J}$, originally introduced by Kaufmann & Koolen (2021),

$$\mathcal{J}(x) := 2\tilde{g}\left(\frac{g^{-1}(1+x) + \ln(\pi^2/3)}{2}\right), \tag{55}$$

where $g^{-1}(y)$ is the inverse function of $g(y) := y - \ln(y)$ defined for $y \geq 1$, and for any $x \geq 0$, $\tilde{g}(x) := e^{1/g^{-1}(x)}g^{-1}(x)$ if $x \geq g^{-1}(1/\ln(3/2))$ and $\tilde{g}(x) = (3/2)(x - \ln(\ln(3/2)))$ otherwise. We select the threshold function

$$\beta(n, \epsilon) := 2\mathcal{J}\left(\frac{\log(3n\sqrt{n}/\epsilon)}{2}\right) + 6\log(1 + \log n), \tag{56}$$

and define $h_i$ in successful detection events $D_i$ to be $h_i := h_i(\alpha, \epsilon)$ with $h_0(\alpha, \epsilon) := 0$ and for $i > 0$,

$$h_i(\alpha, \epsilon) := \left\lceil 2\left(\frac{4}{\left(\delta^{(i)}\right)^2}\beta(T, \epsilon) + 2\right)\left(\frac{K}{2\alpha} + 1\right)\sqrt{s_i + 1} \right.$$
$$\left. + \left(\frac{4}{\left(\delta^{(i)}\right)^2}\beta(T, \epsilon) + 2\right)^2\left(\frac{K}{2\alpha} + 1\right)^2 \right\rceil, \tag{57}$$

which guarantees that with the proposed diminishing exploration, we will observe

$$\left\lceil \frac{4}{\left(\delta^{(i)}\right)^2}\beta\left(\frac{3}{2}s_i, \epsilon\right) + 1 \right\rceil, \tag{58}$$

post-change samples for each $k \in \mathcal{K}$ after $\nu_i$. We analyze the GLR-UCB with diminishing exploration under the following assumption:

**Assumption C.7.** $\nu_i - \nu_{i-1} \geq 2\max\{h_i, h_{i-1}\}$ for all $i \in \{1, \ldots, M\}$.

Following the proof of Lemma 8 in Besson et al. (2022), we can show the following lemma:

**Lemma C.8.** *Under assumption C.7 and Equation 57, it holds that*

$$\sum_{i=1}^{M}\mathbb{P}\left(F_i|\overline{F}_{i-1}D_{i-1}\right) + \sum_{i=1}^{M-1}\mathbb{P}\left(\overline{D}_i|\overline{F}_i\overline{F}_{i-1}D_{i-1}\right) \leq \epsilon(K+1)M. \tag{59}$$

Plugging equation 57 and Lemma C.8 into Theorem 4.1 shows the following Corollary 4.6.

**Corollary 4.6** Combining Algorithm 1 and 4 with $\beta$ function in equation 5 achieves the expected regret upper bound as follows:

$$\mathbb{E}[R(1,T)] \leq \underbrace{\sum_{i=1}^{M}\tilde{C}_i}_{(a)} + \underbrace{2\alpha\sqrt{MT}}_{(b)} + \underbrace{\left(\frac{4}{\left(\delta^{(i)}\right)^2}\beta(T, \epsilon) + 2\right)^2\left(\frac{K}{2\alpha} + 1\right)^2 M}_{(c)}$$
$$+ \underbrace{2\left(\frac{4}{\left(\delta^{(i)}\right)^2}\beta(T, \epsilon) + 2\right)\left(\frac{K}{2\alpha} + 1\right)\sqrt{M(T+M)}}_{(c)} + \underbrace{\epsilon(K+1)M}_{(d)}, \tag{60}$$

where $\tilde{C}_i = 8\sum_{\Delta_k^{(i)} > 0}\frac{\log T}{\Delta_k^{(i)}} + \left(\frac{5}{2} + \frac{\pi^2}{3} + K\right)\sum_{k=1}^{K}\Delta_k^{(i)}$. By setting $\alpha = c\sqrt{K\log(KT)}$ for some constant $c$ and $\epsilon = 1/\sqrt{T}$, the expected regret is upper-bounded by $\mathcal{O}(\sqrt{KMT\log T})$.

### C.2 PROOF OF SECTION 5

This subsection covers the proofs of the extended results discussed in Section 5. These extensions include advanced integrations and new theoretical insights.

**Corollary C.9.** *We can extend Lemma C.1 to the case where we only care about the optimal arm changing to another one. Then, the number of times that arm $k$ is selected in the exploration phase and the diminishing exploration regret during the time interval $[\tau_{r-1}^*, \nu_r^*)$ would be*

$$N_{\mathrm{DE},k}\left(\tau_{r-1}^*, \nu_r^*\right) \le \frac{2\alpha\sqrt{\nu_r^* - \tau_{r-1}^*}}{K} + \frac{3}{2}, \tag{61}$$

*and*

$$\mathbb{E}\left[R_{\mathrm{DE}}\left(\tau_{r-1}^*, \nu_r^*\right)\right] \le 2\alpha\sqrt{\nu_r^* - \tau_{r-1}^*} + \frac{3}{2}K. \tag{62}$$

**Corollary C.10.** *We can extend Lemma C.3 to the case where we only care about the optimal arm changing to another one. Then, the number of times that arm $k$ is selected in the exploration phase and the diminishing exploration regret during $[\tau_{r-1}^*, \nu_r^*)$ can be bounding by*

$$\mathbb{E}\left[R\left(\tau_{r-1}^*, \nu_r^*\right)\Big|\overline{F}_{r-1}^* D_{r-1}^*\right] \le \tilde{C} + 2\alpha\sqrt{s_r^*} + T \cdot \mathbb{P}\left(F_r^*\Big|\overline{F}_{r-1}^* D_{r-1}^*\right), \tag{63}$$

**Lemma C.11.** *The false alarm of the super segment can be bounded as follow:*

$$\mathbb{P}\left(F_r^*\Big|\overline{F}_{r-1}^* D_{r-1}^*\right) \le \sum_{k=1}^K \mathbb{P}\left(\overline{Ignore},\ optimal\ arm\ no\ change\ \Big|arm\ k\ alarm,\ \overline{F}_{r-1}^* D_{r-1}^*\right) \tag{64}$$

*Proof.* Conditioning on $\overline{F}_{r-1}^* D_{r-1}^*$ holds, we have the event

$$F_r^* = \bigcup_{k=1}^K \left\{\overline{Ignore}, \text{optimal arm no change, arm } k \text{ alarm}\right\}. \tag{65}$$

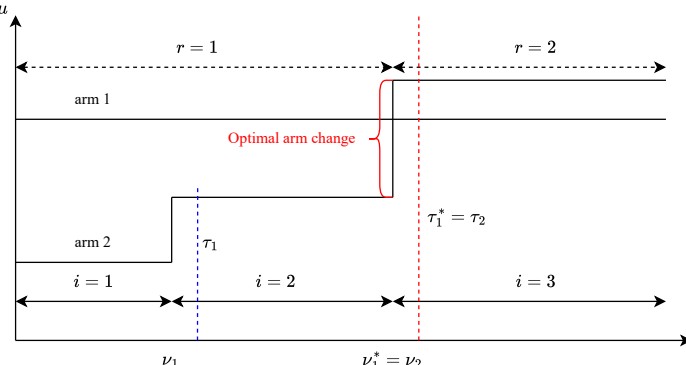

Figure 5: This figure allows us to easily compare the notation in Section 4 and 5.

Using the union bound,

$$\mathbb{P}\left(F_r^*\middle|\overline{F}_{r-1}^* D_{r-1}^*\right) = \mathbb{P}\left(\bigcup_{k=1}^{K}\{\overline{Ignore}, \text{optimal arm no change, arm } k \text{ alarm}\}\middle|\overline{F}_{r-1}^* D_{r-1}^*\right)$$

(66a)

$$\leq \sum_{k=1}^{K} \mathbb{P}\left(\overline{Ignore}, \text{optimal arm no change, arm } k \text{ alarm}\middle|\overline{F}_{r-1}^* D_{r-1}^*\right) \quad (66b)$$

$$= \sum_{k=1}^{K} \mathbb{P}\left(\overline{Ignore}, \text{optimal arm no change}\middle|\text{arm } k \text{ alarm}, \overline{F}_{r-1}^* D_{r-1}^*\right) \quad (66c)$$

$$\times \mathbb{P}\left(\text{arm } k \text{ alarm}\middle|\overline{F}_{r-1}^* D_{r-1}^*\right) \quad (66d)$$

$$\leq \sum_{k=1}^{K} \mathbb{P}\left(\overline{Ignore}, \text{optimal arm no change}\middle|\text{arm } k \text{ alarm}, \overline{F}_{r-1}^* D_{r-1}^*\right) \quad (66e)$$

$\square$

**Lemma C.12.** *The miss detection probability of the super segment can be bounded as follow:*

$$\mathbb{P}\left(\overline{D}_r^*\middle|\overline{F}_r^* \overline{F}_{r-1}^* D_{r-1}^*\right) \leq \sum_{k=1}^{K} \mathbb{P}\left(\overline{D}_{r,k}\middle|\overline{F}_r^* \overline{F}_{r-1}^* D_{r-1}^*\right) \quad (67)$$

*Proof.* We make some modifications to event $D_i$ in Section 4 and extend $D_{r,k}$ as the arm $k$ alarm in the tolerate delay (regardless of whether to ignore or not) after the $r$-th optimal arm change. Conditioning on $\overline{F}_r^* \overline{F}_{r-1}^* D_{r-1}^*$ holds, we have the event

$$\overline{D}_r^* = \bigcup_{k=1}^{K} \overline{D}_{r,k}. \quad (68)$$

Using the union bound, we can get the result as Equation 67. $\square$

**Theorem C.13** (General form of regret bound). *Insert Algorithm 2 into Algorithm 1 and combine with a CD algorithm, which achieves the expected regret upper bound as follows:*

$$\mathbb{E}\left[R\left(1,T\right)\right] \leq \underbrace{\sum_{r=1}^{S} \tilde{C}_r^*}_{(a)} + \underbrace{2\alpha\sqrt{ST}}_{(b)} + \underbrace{\sum_{r=1}^{S-1}\left(\mathbb{E}\left[\tau_r^* - \nu_r^*\middle|D_i^*\overline{F}_i^* D_{i-1}^*\overline{F}_{i-1}^*\right] + d_{I,r}\right)}_{(c)}$$

$$+ \underbrace{T\sum_{r=1}^{S-1}\mathbb{P}\left(F_r^*\middle|\overline{F}_{r-1}^* D_{r-1}^*\right) + \mathbb{P}\left(\overline{D}_r^*\middle|\overline{F}_r^* \overline{F}_{r-1}^* D_{r-1}^*\right)}_{(d)} + \underbrace{K\left(M-1\right)N_I + T\sum_{i=1}^{M}\mathbb{P}\left(F_i\middle|\overline{F}_{i-1} D_{i-1}\right)}_{(e)},$$

(69)

*where $\tilde{C}_r^* = 8\sum_{\min_{\nu_{r-1}^* \leq t \leq \nu_r^*}\Delta_{k,t}>0}\frac{\log T}{\min_{\nu_{r-1}^* \leq t \leq \nu_r^*}\Delta_{k,t}} + \left(\frac{5}{2} + \frac{\pi^2}{3} + K\right)\sum_{k=1}^{K}\max_{\nu_{r-1}^* \leq t \leq \nu_r^*}\Delta_{k,t}$, and term (e) represents the cost incurred during the period of deciding whether to skip, which includes the additional exploration cost for collecting sufficient observations for the skipping mechanism. This cost applies to all the changes detected, including actual changes and false alarms.*

*Moreover, we can transform Equation 69 into another form using lemma C.11 and C.12 as follows:*

$$\mathbb{E}\left[R\left(1,T\right)\right] \leq \underbrace{\sum_{r=1}^{S} \tilde{C}_r^*}_{(a)} + \underbrace{2\alpha\sqrt{ST}}_{(b)} + \underbrace{\sum_{r=1}^{S-1}\left(\mathbb{E}\left[\tau_r^* - \nu_r^* \Big| D_i^* \overline{F}_i^* D_{i-1}^* \overline{F}_{i-1}^*\right] + d_{I,r}\right)}_{(c)}$$

$$+ \underbrace{T\sum_{r=1}^{S-1}\sum_{k=1}^{K}\mathbb{P}\left(\overline{Ignore}, optimal\ arm\ no\ change \Big| arm\ k\ alarm, \overline{F}_{r-1}^* D_{r-1}^*\right) + T\sum_{r=1}^{S-1}\sum_{k=1}^{K}\mathbb{P}\left(\overline{D_{r,k}} \Big| \overline{F}_r^* \overline{F}_{r-1}^* D_{r-1}^*\right)}_{(d)},$$

$$(70)$$

**Lemma C.14.** *Suppose an arm $k \in \mathcal{K}$ changes at time $\nu$ and raises an alarm at time $\tau$, but the optimal arm is the same one, and we choose a variable $N_I$ such that $N_k\left(\nu,\tau\right) + N_I \geq \frac{4\xi\log T}{\Delta_{\min}^2}$ and $\xi = 1$, we have*

$$\mathbb{P}\left(\overline{Ignore}, optimal\ arm\ no\ change \Big| arm\ k\ alarm, \overline{F}_{r-1}^* D_{r-1}^*\right) \leq \frac{2}{T^2} \qquad (71)$$

*Proof.* Condition on arm $k$ alarms, $\overline{F}_{r-1}^*$ and $D_r^*$ hold, the event

$$\left\{\overline{Ignore}, optimal\ arm\ no\ change\right\} \subset \left\{\hat{\mu}_k \geq \mu_k + \sqrt{\frac{\xi\log T}{N_k\left(\nu,\tau\right) + N_I}}\right\} \qquad (72a)$$

$$\cup \left\{\hat{\mu}_{k^*} \leq \mu_{k^*} - \sqrt{\frac{\xi\log T}{N_{k^*}\left(\nu,\tau\right) + N_I}}\right\} \qquad (72b)$$

$$\cup \left\{\mu_{k^*} - \mu_k < 2\sqrt{\frac{\xi\log T}{N_k\left(\nu,\tau\right) + N_I}}, N_k\left(\nu,\tau\right) + N_I \geq \frac{4\xi\log T}{\Delta_{\min}^2}\right\}. \qquad (72c)$$

The third event will vanish because

$$2\sqrt{\frac{\xi\log T}{N_k\left(\nu,\tau\right) + N_I}} \leq 2\sqrt{\frac{\xi\log T \Delta_{\min}^2}{4\xi\log T}} = \Delta_{\min} \leq \mu_{k^*} - \mu_k \qquad (73)$$

Equation 73 substitutes the latter term of event 72c into the former term. As a result of the substitution, it is determined that this event cannot occur, hence the probability is zero. Therefore, we only need to consider events 72a and 72b. Using the Chernoff-Hoeffding bound, we can obtain

$$\mathbb{P}\left(\hat{\mu}_k \geq \mu_k + \sqrt{\frac{\xi\log T}{N_k\left(\nu,\tau\right) + N_I}}\right) \leq T^{-2\xi}. \qquad (74)$$

$$\mathbb{P}\left(\hat{\mu}_{k^*} \leq \mu_{k^*} - \sqrt{\frac{\xi\log T}{N_{k^*}\left(\nu,\tau\right) + N_I}}\right) \leq T^{-2\xi} \qquad (75)$$

If we choose $\xi = 1$, then

$$\mathbb{P}\left(\hat{\mu}_k \geq \mu_k + \sqrt{\frac{\xi\log T}{N_k\left(\nu,\tau\right) + N_I}}\right) + \mathbb{P}\left(\hat{\mu}_{k^*} \leq \mu_{k^*} - \sqrt{\frac{\xi\log T}{N_{k^*}\left(\nu,\tau\right) + N_I}}\right) \leq \frac{2}{T^2} \qquad (76a)$$

$\square$

### C.2.1 PROOF OF INTEGRATION WITH CHANGE DETECTORS OF M-UCB

**Assumption C.15.** The algorithm knows a lower bound $\delta > 0$ such that $\delta \leq \min_i \max_{k \in \mathcal{K}} \delta_k^{(i)}$.

**Assumption C.16.** $s_r^* = \Omega\left(\left(\log KT + \sqrt{K\log KT}\right)\sqrt{s_{r-1}^*}\right)$.

In particular, if $s_r^* = \Theta\left(\left(\log KT + \sqrt{K \log KT}\right)^{2(1+\epsilon)}\right)$ for every $i$, Assumption C.16 holds.

**Corollary 5.2** Combining Algorithm 1 and 3 with the parameters in Equation 6, and Equation 4 achieves the expected regret upper bound as follows:

$$\mathbb{E}\left[R\left(1,T\right)\right] \leq \underbrace{\sum_{r=1}^{S} \tilde{C}_r^*}_{(a)} + \underbrace{2\alpha\sqrt{ST}}_{(b)} + \underbrace{w\left(\frac{K}{2\alpha}+1\right)\sqrt{S\left(T+S\right)}}_{(c)}$$

$$+ \underbrace{\frac{w^2 S}{4}\left(\frac{K}{2\alpha}+1\right)^2}_{(c)} + \underbrace{2S}_{(d)}, \quad (77)$$

where
$\tilde{C}_r^* = 8\sum_{\min_{\nu_{r-1}^* \leq t \leq \nu_r^*} \Delta_{k,t} > 0} \frac{\log T}{\min_{\nu_{r-1}^* \leq t \leq \nu_r^*} \Delta_{k,t}} + \left(\frac{5}{2}+\frac{\pi^2}{3}+K\right)\sum_{k=1}^{K}\max_{\nu_{r-1}^* \leq t \leq \nu_r^*} \Delta_{k,t}$. By
setting $\alpha = c\sqrt{K \log\left(KT\right)}$ for some constant $c$, the expected regret is upper-bounded by
$\mathcal{O}(\sqrt{KST \log T})$.

*Proof.* We can substitute lemma C.6, lemma C.4, and lemma C.5 into terms (c) and (d) of Equation 70 respectively, and term (e) could be zero because $w$ (window size) samples are sufficient to determine whether to ignore. We don't need to make an additional decision interval to ensure that the number of samples is sufficient for ignoring. $\square$

### C.2.2 PROOF OF INTEGRATION WITH CHANGE DETECTORS OF GLR-UCB

**Corollary 5.3** Combining Algorithm 1 and 4 with $\beta$ function in Equation 5 achieves the expected regret upper bound as follows:

$$\mathbb{E}\left[R\left(1,T\right)\right] \leq \underbrace{\sum_{r=1}^{S} \tilde{C}_r^*}_{(a)} + \underbrace{2\alpha\sqrt{ST}}_{(b)} + \underbrace{\frac{2KS}{T}+\epsilon S(K+1)}_{(d)} + \underbrace{K(M-1)\frac{4\log T}{\Delta_{\min}^2}+\epsilon MK}_{(e)}$$

$$+ \underbrace{\left(\frac{4}{\delta^2}\beta\left(T,\epsilon\right)+2\right)^2\left(\frac{K}{2\alpha}+1\right)^2 S + 4\left(\frac{2}{\delta^2}\beta\left(T,\epsilon\right)+1\right)\left(\frac{K}{2\alpha}+1\right)\sqrt{S\left(T+S\right)}}_{(c)}, \quad (78)$$

where
$\tilde{C}_r^* = 8\sum_{\min_{\nu_{r-1}^* \leq t \leq \nu_r^*} \Delta_{k,t} > 0} \frac{\log T}{\min_{\nu_{r-1}^* \leq t \leq \nu_r^*} \Delta_{k,t}} + \left(\frac{5}{2}+\frac{\pi^2}{3}+K\right)\sum_{k=1}^{K}\max_{\nu_{r-1}^* \leq t \leq \nu_r^*} \Delta_{k,t}$. By
setting $\alpha = c\sqrt{K \log\left(KT\right)}$ for some constant $c$ and $\epsilon = 1/\sqrt{T}$, the expected regret is
upper-bounded by $\mathcal{O}(\sqrt{KST \log T})$.

*Proof.* We can substitute Equation 57, and lemma C.8 into terms (c) and (d) of Equation 70 respectively.

$\square$

## D ALGORITHMS AND PARAMETERS TUNING

In this appendix, we provide an explanation of our parameter selection. For M-UCB, the window size $w$ is set to 200 unless otherwise specified; however, for the last data point ($M = 100$) in Figure 3b, we chose $w = 50$ due to the limitations inherent to change detection. We compute the change detection threshold $b_{\text{M-UCB}} = \sqrt{w/2 \log\left(2KT^2\right)}$ following the original formulation in Cao et al. (2019). Additionally, the uniform exploration rate $\gamma_{\text{M-UCB}} = \sqrt{MK \log T / T}$ is determined as initially stated in Besson et al. (2022).Concerning CUSUM-UCB, we adhere to Liu et al. (2018) by

fixing $\epsilon = 0.1$, setting the change detection threshold $b_{\text{CUSUM-UCB}} = \log{(T/M - 1)}$, and establishing the uniform exploration rate $\gamma_{\text{CUSUM-UCB}} = \sqrt{MK\log T/T}$ as initially stated in Besson et al. (2022). Additionally, in CUSUM-UCB, the change point detection involves averaging the first $H$ samples, where $H$ is set to 100. For GLR-UCB (Besson et al., 2022), we set $\gamma_{m,\text{GLR-UCB}} = \sqrt{mK\log T/T}$, where $m$ is the number of alarms. We utilize the threshold function $\beta(n, \delta) = \log{\left(n^{3/2}/\delta\right)}$ and set $\delta = 1/\sqrt{T}$. In our setup, for both the diminishing versions of M-UCB and CUSUM-UCB, we follow the parameter selection approach described earlier, except for the choice of the exploration rate. In this context, we opt for $\alpha = 1$. For passive methods, including DUCB (Garivier & Moulines, 2011) and DTS (Qi et al., 2023), we use a discounting factor $\gamma = 0.75$. MASTER, on the other hand, follows the theoretical settings outlined in Wei & Luo (2021) and is categorized as an active method.

To evaluate the scalability of our methods, we conducted three sets of scaling experiments. For scaling in $t$ (Figure 3a), scaling in $M$ (Figure 3b), and scaling in $T$ (Figure 3c), The parameters of DUCB, DTS and MASTER follow the theoretical settings outlined in Garivier & Moulines (2011), Qi et al. (2023) and Wei & Luo (2021), respectively.

Table 2: Parameter Selection for Active and Passive Methods

| Method | Parameters | References |
|---|---|---|
| **M-UCB** | - Window size $w = 200$ (default), $w = 50$ for $M = 100$ (Figure 3b). 
 - Threshold: $b_{\text{M-UCB}} = \sqrt{w/2\log{(2KT^2)}}$. 
 - Exploration rate: $\gamma_{\text{M-UCB}} = \sqrt{MK\log T/T}$. | Cao et al. (2019), |
| **CUSUM-UCB** | - Fixed $\epsilon = 0.1$. 
 - Threshold: $b_{\text{CUSUM-UCB}} = \log{(T/M - 1)}$. 
 - Exploration rate: $\gamma_{\text{CUSUM-UCB}} = \sqrt{MK\log T/T}$. 
 - Change detection based on averaging first $H = 100$ samples. | Liu et al. (2018), |
| **GLR-UCB** | - Exploration rate: $\gamma_{m,\text{GLR-UCB}} = \sqrt{mK\log T/T}$. 
 - Threshold function: $\beta(n, \delta) = \log{\left(n^{3/2}/\delta\right)}$, $\delta = 1/\sqrt{T}$. | Besson et al. (2022) |
| **MASTER** | - All parameters follow theoretical settings. | Wei & Luo (2021) |
| **DUCB (Passive)** | - Discounting factor $\gamma = 0.75$.(Figure 3a and 3d) 
 - Discounting factor follows theoretical setting. (Figure 3b and 3c) | Garivier & Moulines (2011) |
| **DTS (Passive)** | - Discounting factor $\gamma = 0.75$. (Figure 3a and 3d) 
 - Discounting factor follows theoretical setting. (Figure 3b and 3c) | Qi et al. (2023) |

## E  ADDITIONAL RELATED WORK

**Structured Non-Stationary Bandits**. Another related line of research works is non-stationary bandit with structured reward changes, which are typically motivated by the dynamic behavior of real-world applications. For example, Heidari et al. (2016) proposes the *Rising bandit* problem, where the rewards are assumed to be a non-decreasing and concave function of the current time index and the number of pulls. Subsequently, this model is extended to the stochastic setting by (Metelli et al., 2022). Another related setting is the *Rotting bandit* (Levine et al., 2017), where the expected reward is a non-increasing function of the number of pulls. Moreover, Zhou et al. (2021) studies the regime switching bandit, where the rewards are jointly controlled by an underlying finite-state Markov chain. However, the algorithms tailored to the above customized formulations are not directly applicable to the general piecewise-stationary MAB problem.

**Bandit Quickest Change Detection.** Since the seminal works (Page, 1954; Lorden, 1971), the quickest change detection (QCD) problem, which involves identifying the change of distribution at an unknown time with minimal delay, has been a well studied detection problem of stochastic processes (Veeravalli & Banerjee, 2014). Bandit QCD, a variant of QCD problem recently proposed by (Gopalan et al., 2021), adds another layer of complexity to the conventional QCD by considering bandit feedback. A concurrent work (Xu et al., 2021) also studies a similar setting, namely multi-stream QCD under sampling control, and proposes a myopic sampling policy that achieves a second-order asymptotically optimal detection delay. Despite the above bandit QCD methods focusing mainly on achieving low detection delay rather than characterizing regret bounds, these recent progress could nicely complement the studies of piecewise stationary bandits.

# F  ADDITIONAL SIMULATIONS

**Regret Scaling in** $K$**.** We considered an environment with $T = 20000$ and $M = 5$ for various $K$, aiming to showcase dynamic regrets versus different $K$. In this experiment, expected rewards are generated randomly. Specifically, we randomly generated 5 instances, averaging each instance over 50 times. Figure 6a demonstrates that our method is not limited to working only in simple environments with small $K$ values but is adaptable to a broader range of scenarios.

**Comepare to AdSwitch, ArmSwitch and Meta Algorithm.** These algorithms are indeed computationally quite complex, evidenced by the time complexity of $O(KT^4)$ in Auer et al. (2019), that of $O(K^2T^2)$ in Abbasi-Yadkori et al. (2023), and the fact that recursive calls of the base algorithm are needed by the algorithm in Suk & Kpotufe (2023). Besides, while being able to achieve near-optimal regret bound asymptotically, these elimination-based algorithms generally would not perform well when the time horizon $T$ is small, shown as Figure 6b and 6c. Figure 6d compares our extension, which incorporates a skipping mechanism, with ArmSwitch, where the latter focuses on tracking the most significant arm switches. For our setup, we defined $\mu_1^{(i)} = 0.8, 0.2$ for $i$ where $(i + 1) \bmod 4 = \{2, 3\}, \{0, 1\}$, and $\mu_2^{(i)} = 0.4, 0.6$ for $i$ where $(i + 2) \bmod 2 = 0, 1$, as well as $\mu_2^{(i)} = 0.4, 0.6$ for $i$ where $(i + 3) \bmod 2 = 0, 1$. In our parameter settings, we set $N_I = 50$ and $\alpha = 1$. The results clearly show that our performance significantly exceeds that of the ArmSwitch. The above discussion precisely constitutes the main reason we did not include these algorithms in our experimental comparison, as such algorithms with $T = 20000 \sim 100000$ would take too long to finish while showing results for small $T$ may look unfair for those excellent algorithms.

**Figure 7.** To explore the applicability of our approach in more general nonstationary settings, we further conducted experiments in a slowly changing environment, where the two arms followed sinusoidal oscillations as considered in (Qi et al., 2023). This setup represents a gradual, continuous shift, distinct from the abrupt changes in our primary focus.

Performance of Active and Passive Methods: Surprisingly, some active methods perform reasonably well in this setting. However, their effectiveness is highly dependent on the choice of change detector. For example, M-UCB struggles in this environment, as it is not designed for smooth transitions. Among the passive methods (e.g., Discounted-klUCB and Discounted-klUCB-TS), setting a discount factor of 0.99 yields strong results, emphasizing their adaptability to slowly changing environments.

**Figure 8.** Our empirical results further support the importance of the skipping mechanism. As illustrated in figure 8, the inclusion of the skipping mechanism significantly improves the performance of the algorithm in regret. By applying the proposed diminishing exploration and the skipping mechanism, we can improve the empirical regrets of various active methods, including M-UCB, CUSUM-UCB, and GLR-UCB.

**Figure 9.** We retain the mean reward structure in figure 3d but introduce controlled variations with an even larger horizon $T = 54000$. These experiments reveal a clear advantage of our DE framework, particularly in more dynamic scenarios.

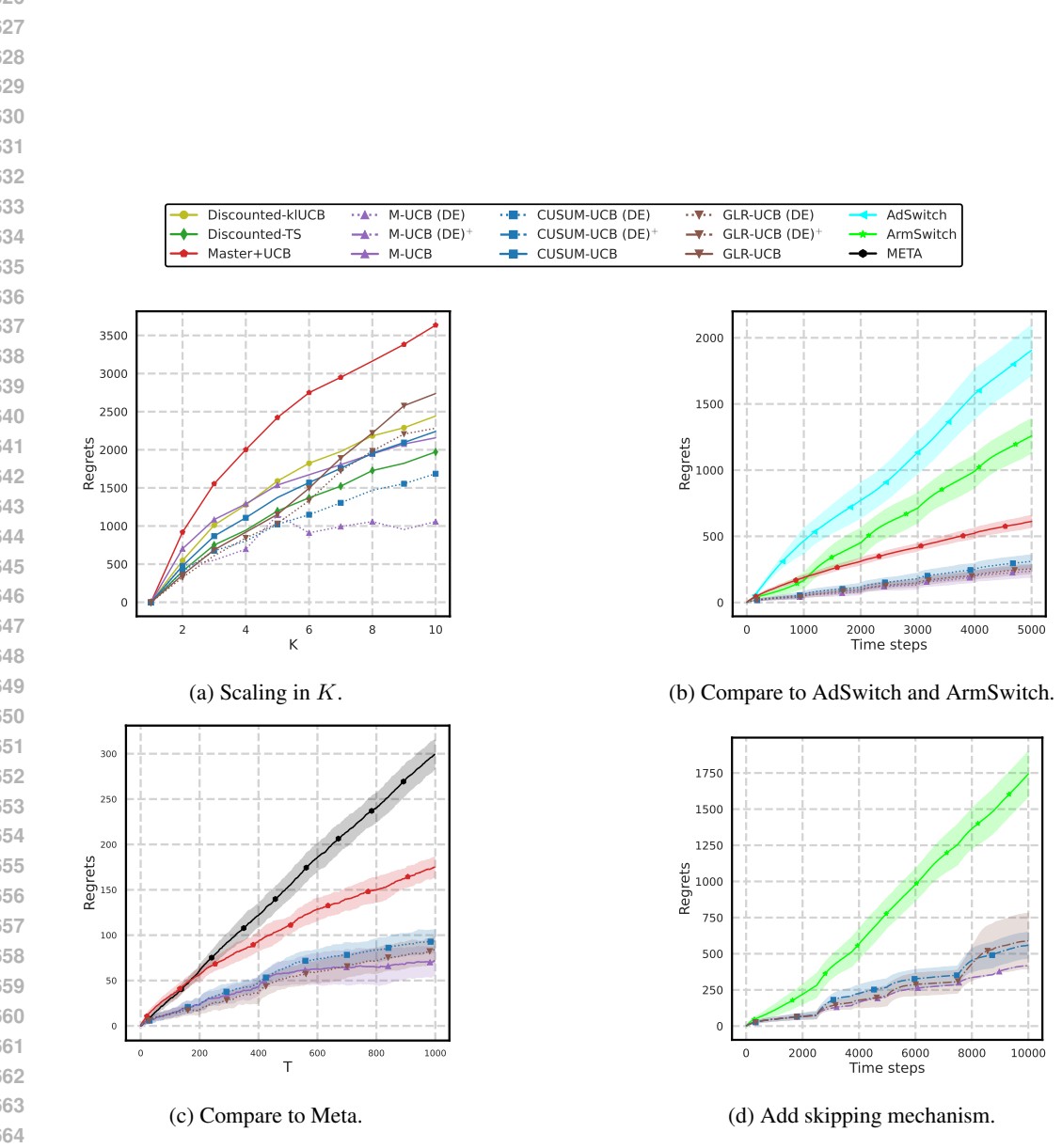

Figure 6: Regret in synthetic environment and yahoo data set.

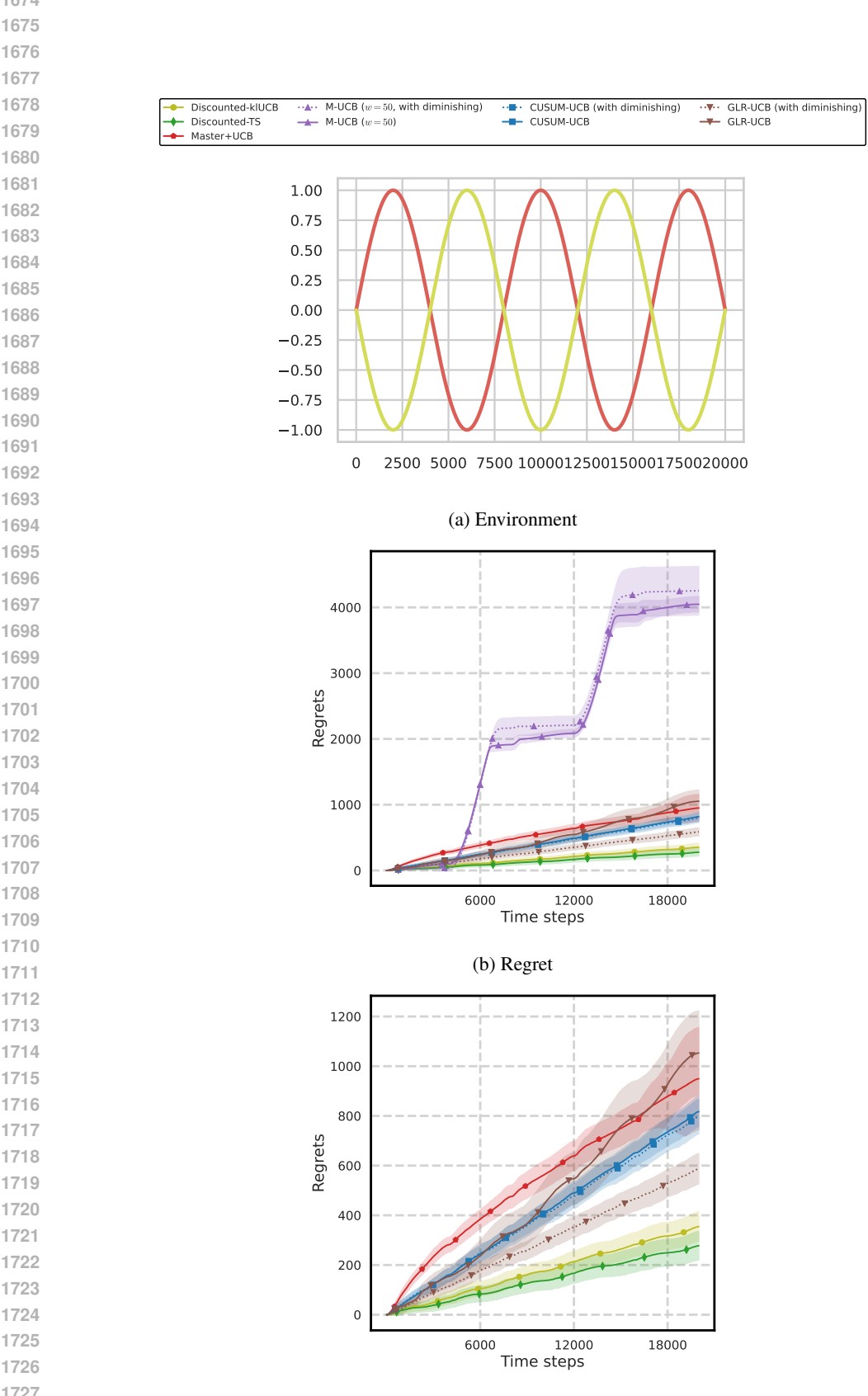

(a) Environment

(b) Regret

(c) A Zoom in version of regret.

Figure 7: The slowly changing environment.

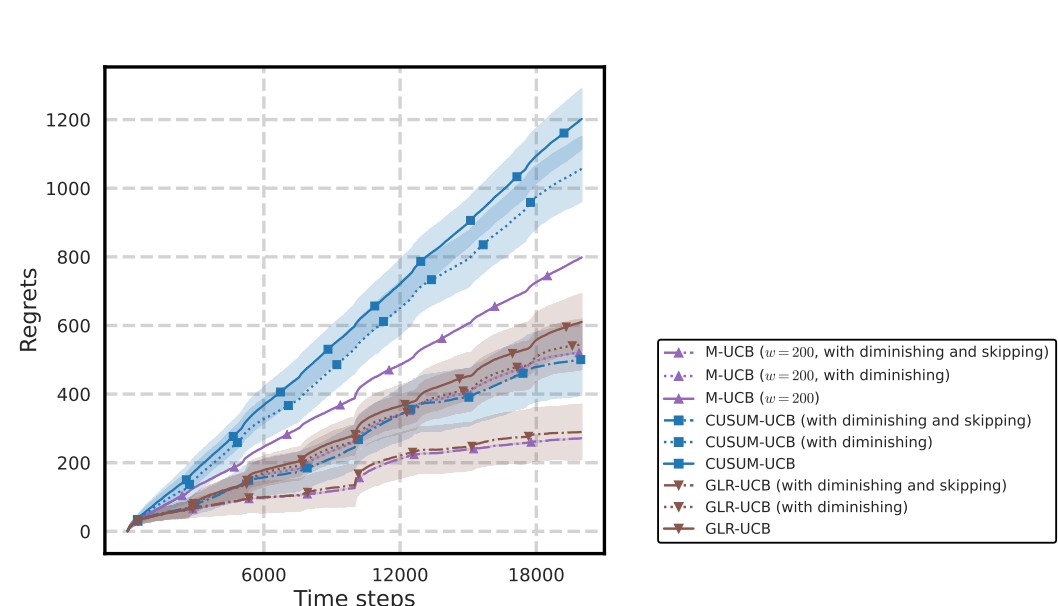

Figure 8: Regret with Skipping Mechanism

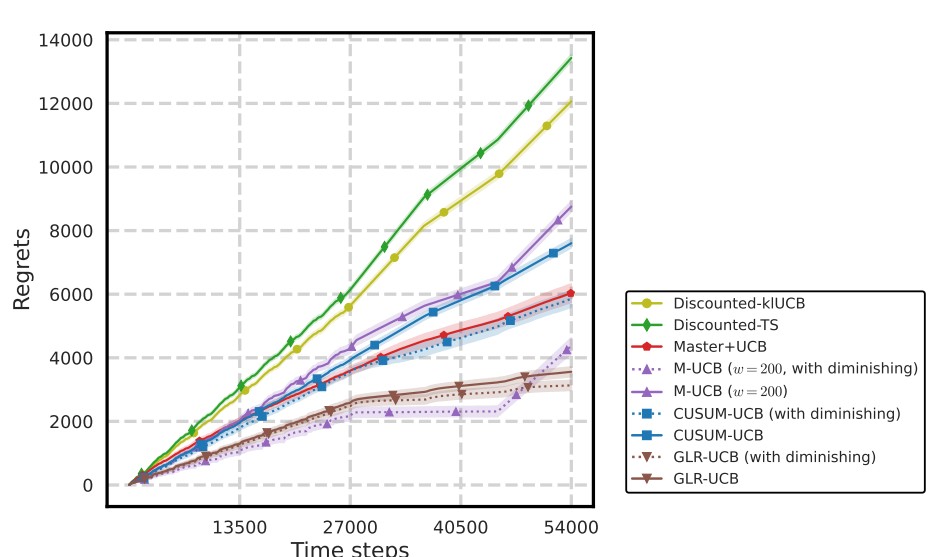

Figure 9: Regret of Yahoo Dataset with larger $M$ and $T$

## G   A ZOOMED-IN VIEW OF FIGURES

Below are the zoomed-in versions of Figure 3 and Figure 4.

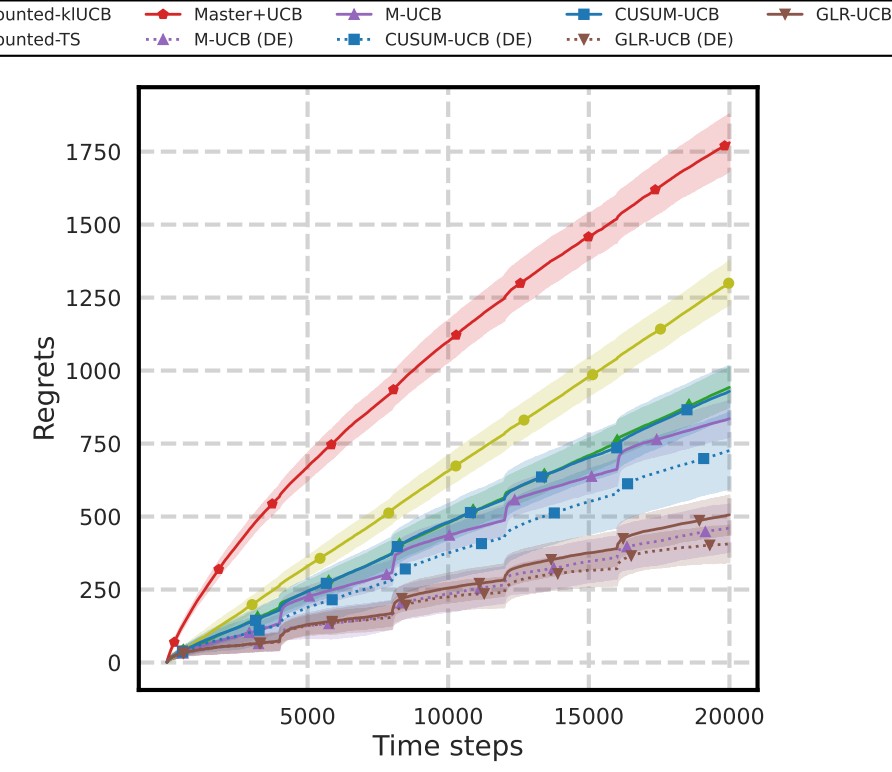

Figure 10: Regret scaling in $t$.

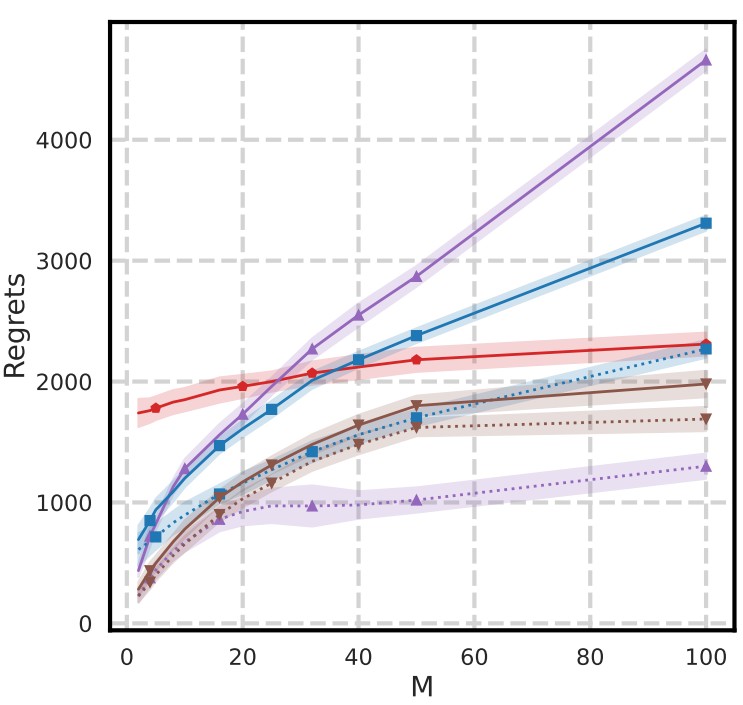

Figure 11: Regret scaling in $M$.

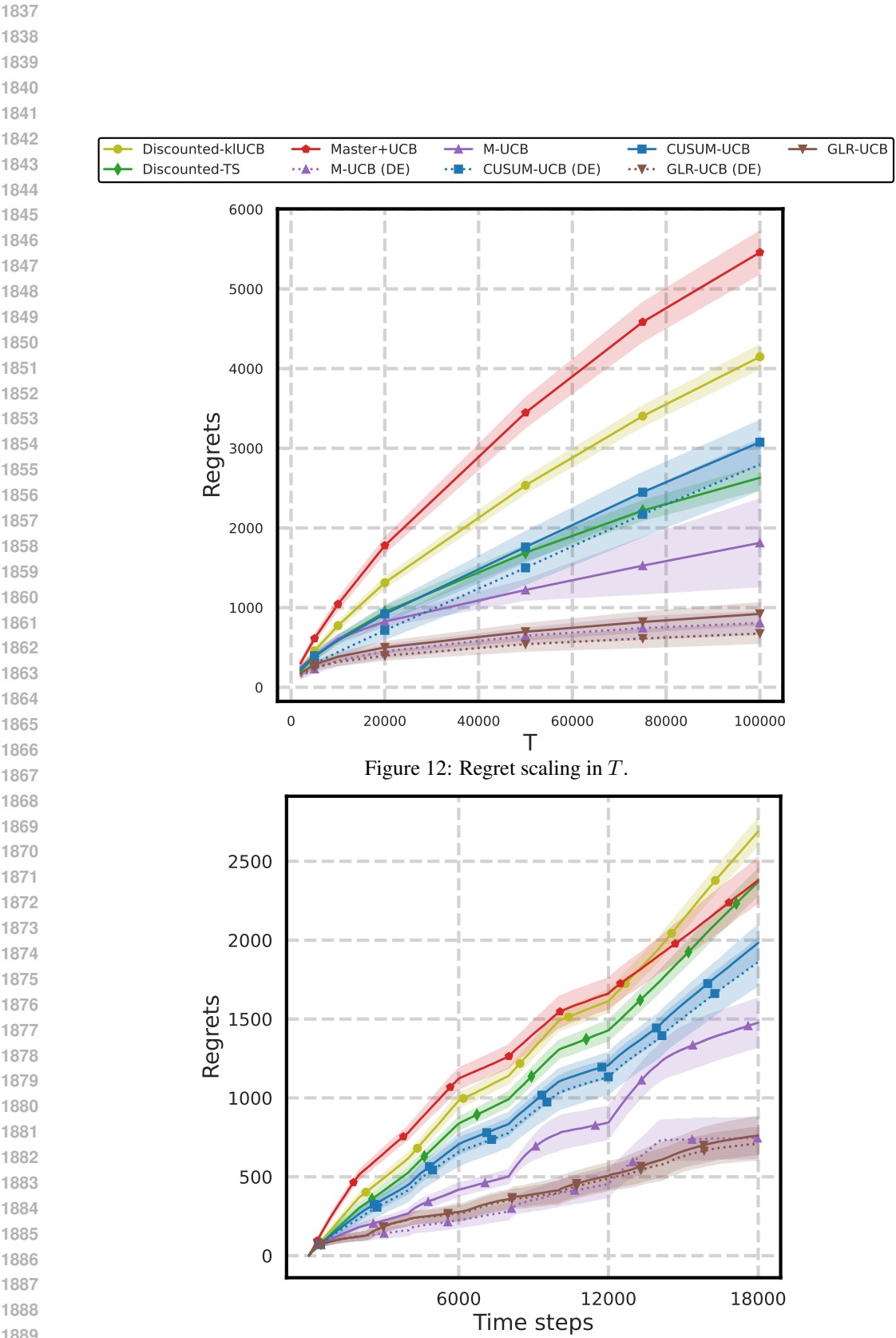

Figure 12: Regret scaling in $T$.

Figure 13: Regret on the Yahoo Dataset.

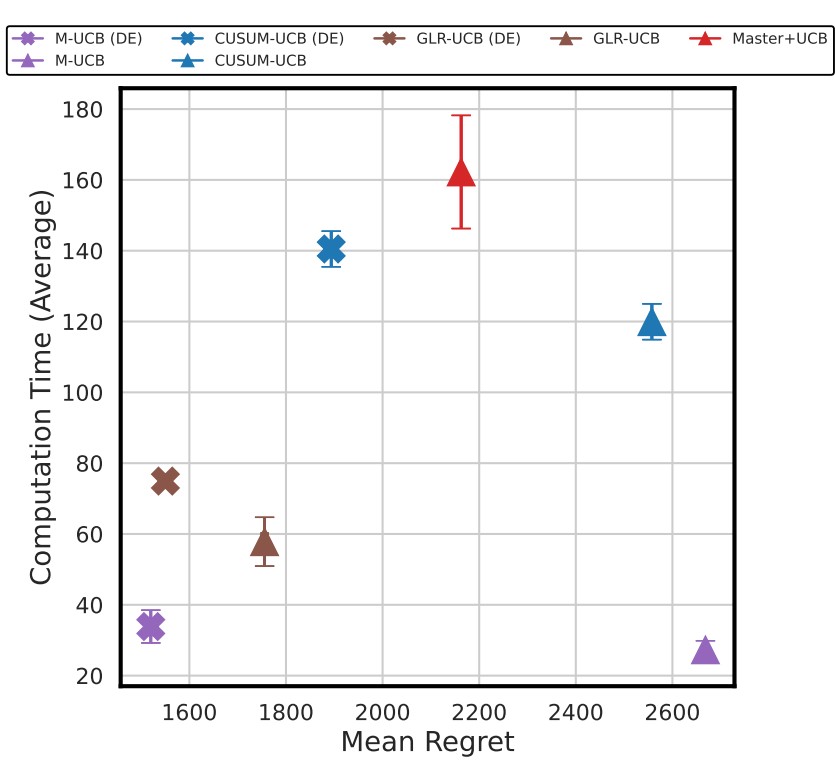

Figure 14: $M = 50, T = 20000$.

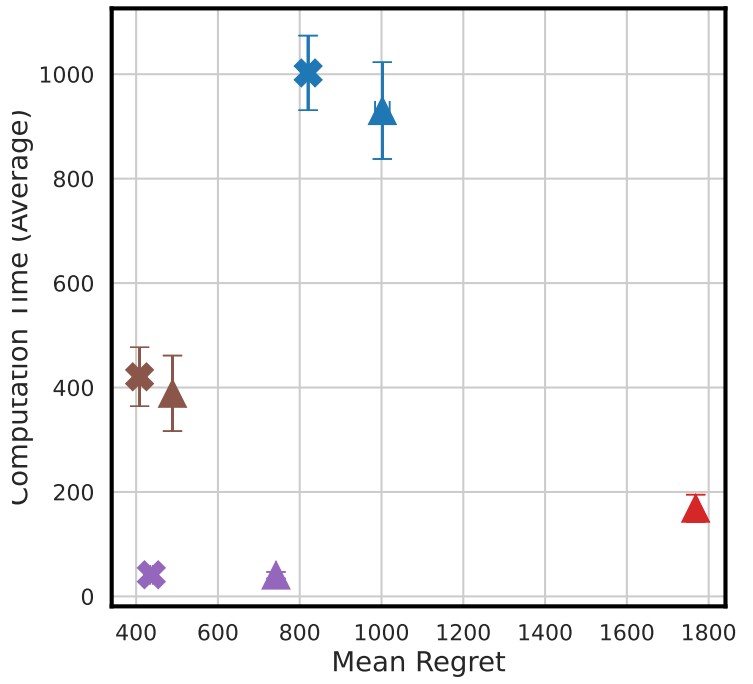

Figure 15: $M = 5, T = 20000$.

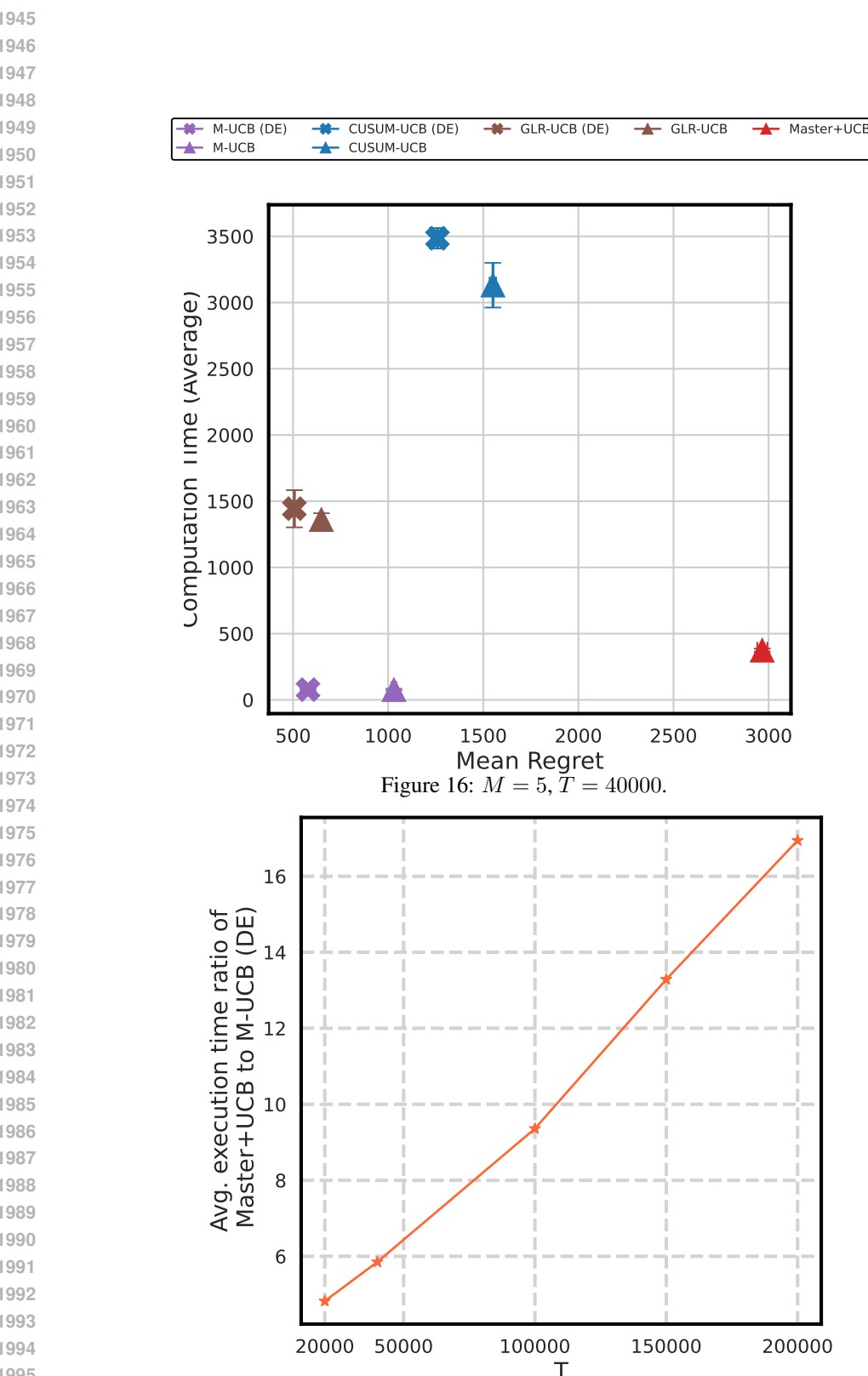

Figure 16: $M = 5, T = 40000$.

Figure 17: Ratio of avg. execution time of Master+UCB to that of M-UCB (DE).

