# OpenReview forum: "Diminishing Exploration: A Minimalist Approach to Piecewise Stationary Multi-Armed Bandits"
_ICLR.cc/2025/Conference — Submitted to ICLR 2025_

### Official Review · Reviewer_zGn5 · 2024-10-17

**Soundness:** 2
**Presentation:** 2
**Contribution:** 1
**Rating:** 3
**Confidence:** 4

**Summary:**

The paper targets piece-wise stationary bandit setting. In this paper, a diminishing exploration mechanism is proposed. Apparently, the mechanism does not require the prior knowledge of the number of stationary segments. This mechanism can be integrated into those algorithms that rely on change point detectors. The authors claim that by using their mechanism in optimizing the forced exploration, they achieve better empirical regret than the traditional forced exploration methods.

**Strengths:**

Their work is authentic and original.

**Weaknesses:**

1. Usually in these kind of settings, there is an assumption that sets a minimum distance between any two consecutive change points in the environment. This type of assumption exists to make sure that the change-point-detector-based algorithm have (statistically) enough time to detect any possible change in the environment and hence the algorithm is able to detect any change to the environment. Naturally, this implies that, as we get more distance from a change point in the environment, the probability of facing another change in the environment is more than the time when we were closer to the last change point. Hence we need to collect more samples from various arms to see if there has been a change that makes our current decisions sub-optimal. This argument (which i believe is correct) has contradiction with the nature of your method in decreasing forced exploration as we get distance from the last change point.

2. The experiments section needs to be clarified more by providing more clear information about the setups.

3. The number of arms involved in the synthetic data experiment is low and it raises the question about the performance of the proposed technique in scenarios with more arms.

4. The writing of the text is not smooth to follow. It can definitely be improved.

**Questions:**

The effect of the mechanism is not motivated within the context of real-life applications. For instance, the question i can ask is; in which application scenario reducing this forced exploration can play an important role? you might want to point to applications with large number of arms. Then I can ask; then why do the experiments focus on small number of arms and do not show the performance of the proposed method in scenarios with a large number of arms?

---

> ### Author Response · Authors · 2024-11-23
>
> # Question 1:
> Usually in these kind of settings, there is an assumption that sets a minimum distance between any two consecutive change points in the environment. This type of assumption exists to make sure that the change-point-detector-based algorithm have (statistically) enough time to detect any possible change in the environment and hence the algorithm is able to detect any change to the environment. Naturally, this implies that, as we get more distance from a change point in the environment, the probability of facing another change in the environment is more than the time when we were closer to the last change point. Hence we need to collect more samples from various arms to see if there has been a change that makes our current decisions sub-optimal. This argument (which i believe is correct) has contradiction with the nature of your method in decreasing forced exploration as we get distance from the last change point.
>
> ## Ans:
> Thank you for your thoughtful feedback! While increasing the exploration rate as time progresses may seem intuitive, this approach struggles in environments with long segments. Without knowing the number of change points ($M$) or segment lengths, exploration can grow excessively large, wasting valuable resources.
>
> Our diminishing exploration strategy provides an alternative that, while counterintuitive at first glance, is provably capable of achieving a perfect balance. In short segments, the high initial exploration rate ensures rapid adaptation to frequent changes. In longer segments, the exploration rate tapers off, minimizing unnecessary exploration while keeping cumulative regret manageable. This dynamic adjustment balances exploration cost and detection delay, adapting naturally to the environment without requiring prior knowledge of $M$.
>
> # Question 2:
> The experiments section needs to be clarified more by providing more clear information about the setups.
>
> ## Ans:
> Thank you for pointing out the need for greater clarity in the experiments section. Upon review, we note that the environment setups for our experiments are described in Section 6, while the algorithm parameters are detailed in Appendix D. However, we acknowledge that some parameter descriptions may be missing or insufficiently detailed. We now list them at the end of this reply.
> To address this, we will explicitly include these missing details in the final version and ensure all parameter settings are clearly presented. Additionally, we recognize that presenting this information in a more structured format, such as a table, would improve clarity and accessibility. We will incorporate such a table in the final version to make the setups and configurations more transparent.
> We appreciate your feedback, as it highlights an important area for improvement, and we are committed to making these adjustments to enhance the clarity of our presentation.
> Parameter settings for Discounted Thompson Sampling (DTS), Discounted UCB (DUCB), and MASTER are as follows:
> ### Scaling in $t$ [Figure 3a]:
> - DUCB:
>     - Discounting factor $\gamma=0.75$
> - DTS:
>     - Discounting factor $\gamma=0.75$
> - MASTER:
>     - All the parameters follow the theoretical setting. (Wei & Luo, 2021)
> ### Scaling in $M$ [Figure 3b]::
> - DUCB:
>     - Discounting factor follows the theoretical setting. (Kocsis & Szepesvári, 2006)
> - DTS:
>     - Discounting factor follows the theoretical setting. (Qi et al., 2023)
> - MASTER:
>     - All the parameters follow the theoretical setting. (Wei & Luo, 2021)
> ### Scaling in $T$ [Figure 3c]:
> - DUCB:
>     - Discounting factor follows the theoretical setting. (Kocsis & Szepesvári, 2006)
> - DTS:
>     - Discounting factor follows the theoretical setting. (Qi et al., 2023)
> - MASTER:
>     - All the parameters follow the theoretical setting. (Wei & Luo, 2021)

---

> > ### Author Response · Authors · 2024-11-23
> >
> > # Question 3:
> > The number of arms involved in the synthetic data experiment is low and it raises the question about the performance of the proposed technique in scenarios with more arms.
> >
> > ## Ans:
> > While our synthetic data experiments involve a small number of arms, we demonstrated in Figure 3d (Yahoo dataset) that our proposed approach performs well even with $K = 6$ in a real-world setting. This result highlights the robustness of our method, even in scenarios with a moderate number of arms, alleviating concerns about its scalability as $K$ increases.
> > Additionally, in Appendix F (Figure 6a), we presented experiments explicitly focused on $K$-scaling (up to $K=10$). In this setup, for each $K$, we randomly generate five sets of reward means, average the results over 50 repetitions per set, and calculate the overall mean performance. These experiments show that our method consistently outperforms standard active methods as $K$ increases, demonstrating its scalability and effectiveness in handling environments with larger numbers of arms.
> > Together, these results provide strong evidence that our approach maintains its performance and computational efficiency as $K$ grows, making it a reliable choice for both synthetic and real-world scenarios with varying numbers of arms.
> >
> >
> > # Question 4:
> > The writing of the text is not smooth to follow. It can definitely be improved.
> >
> > ## Ans:
> > Thank you for your valuable feedback regarding the clarity of the writing. We sincerely apologize if certain parts of the text were not as smooth or easy to follow as intended. We will carefully review the manuscript and make significant improvements to the structure, flow, and readability of the text to ensure that our ideas are communicated more clearly.
> > Your feedback is greatly appreciated, and we are committed to revising the writing to meet the highest standards of clarity and precision.

---

> ### Author Response · Authors · 2024-11-27
>
> We noticed that the reviewer has changed the rating to 3. We would greatly appreciate it if you could provide further explanation regarding this change, as it would help us better understand and address any remaining concerns.
> Additionally, we realized that we may have overlooked one of your earlier comments regarding the practical application and relevance of our approach. Thank you for bringing this up, and we would like to provide clarification here:
>
>
> # Question
> The effect of the mechanism is not motivated within the context of real-life applications. For instance, the question i can ask is; in which application scenario reducing this forced exploration can play an important role? you might want to point to applications with large number of arms. Then I can ask; then why do the experiments focus on small number of arms and do not show the performance of the proposed method in scenarios with a large number of arms?
>
> # Answer
> ## The Advantage of DE in Unknown $M$ Settings:
> The key strength of diminishing exploration (DE) lies in its ability to operate effectively without requiring prior knowledge of $M$, the number of change points. Many traditional methods depend on knowing $M$ to set appropriate exploration rates, but this is often impractical in real-world applications where $M$ is difficult to estimate or entirely unknown.
> ## Targeting Practical Applications Without $M$ :
> Our method is particularly well-suited for scenarios where $M$ is unknown, which covers almost every practical scenarios. In environments with dynamic changes, such as recommendation systems, online advertising, or pricing strategies, $M$ is rarely accessible, making DE a highly valuable and versatile approach.
> ## Performance in Scenarios with Large Number of Arms:
> Our method is not particularly vulnerable to scenarios with a large number of arms. In fact, our simulations demonstrate this adaptability. Regarding the comment about the simulations about small number of arms, we believe that this is a misunderstanding. For example, in the Yahoo dataset experiments (Figure 3d), we show that DE maintains robust performance even with six arms in a real-world dataset. Furthermore, in $K$-scaling experiments (Appendix F, Figure 6a), our results reveal that our method consistently outperforms standard active methods as $K$ increases. This highlights its strong scalability and effectiveness in handling scenarios with a larger number of arms.
>
> Therefore, while the computational complexity of our method remains unchanged, the adaptability to unknown $M$ and robust performance in both small- and large-arm settings make DE a highly practical and efficient framework for real-world applications. We hope this clarification addresses your concerns and highlights the broad applicability of our approach.

---

> ### Author Response · Authors · 2024-12-01
>
> We wanted to kindly follow up on the rebuttal we submitted. We deeply value your feedback and would greatly appreciate any additional insights into this change. Understanding your perspective would help us address any remaining concerns more effectively, especially given that you have changed the grade to a 3.
> If there are areas that require further clarification or additional information, we would be happy to provide it. Thank you once again for your time and thoughtful comments. We truly value your efforts and appreciate your continued guidance throughout the review process.

---

### Official Review · Reviewer_o4hn · 2024-11-01

**Soundness:** 3
**Presentation:** 3
**Contribution:** 2
**Rating:** 5
**Confidence:** 3

**Summary:**

This paper considers the non-stationary stochastic bandit problem, where the reward distributions of arms can change over time. The challenge is to balance the trade-off between variations in these distributions and following a bandit algorithm to optimize the current reward at each stage. To this end, the paper proposes a new change detection algorithm using a diminishing exploration rate, which can be coupled with existing change detection frameworks and bandit algorithms. The paper provides a theoretical guarantee for the proposed algorithm, demonstrated by an optimal regret guarantee. Additionally, it extends the problem to a specific setting involving arm changes and shows a regret guarantee for this case as well. Lastly, a numerical study with simulation experiments demonstrates significant improvement over existing algorithms.

**Strengths:**

1. The paper is clearly written and easy to follow. The comparison with existing work in Table 1 is particularly helpful in understanding the contributions of this paper.
2. The proposed diminishing rate is novel in this specific non-stationary bandit setting. In addition, the algorithm does not require knowledge of the total variation and is relatively easy to implement.
3. The claims are well-supported by the theoretical guarantees. Clear explanations of the theorems are provided, aiding in understanding the key takeaways.
4. Extensive numerical experiments are conducted to demonstrate the algorithm’s performance, showing significantly smaller regret compared to existing benchmarks.

**Weaknesses:**

1. If I understand this paper correctly, one of the main contributions of the paper is that the proposed algorithm does not require knowledge of $M$ and achieves a $\sqrt{MT}$ guarantee and $KT$ computation complexity. One of my major concerns is that, based on Table 1, it seems like (Wei & Luo 2021) has studied the setting without knowledge of $M$, proposing the algorithm Master, which also achieves optimal regret and the same computational complexity. So I am a bit confused by the paper's claim that it studies this novel setting with an optimal guarantee.

2. Additionally, aside from this single parameter change (exploration rate), it would be helpful to understand what other novelties are introduced. Based on the current modification, it is somewhat unclear how the algorithm stands out in terms of novelty.

3. The second concern is about the choice of $\alpha$. It seems like the paper never explicitly states how to choose the parameter $\alpha$, which, however, is key in the algorithm and used in almost all cases. Also, the regret bound has a dependency on $\alpha$. Therefore, without this, it is hard to quantify how we can deploy this algorithm without parameter tuning and how the regret really changes with $T$.

4. The paper lists the statement about the computational complexity in Table 1, which is $KT$. However, it seems like there is no theoretical or formal statement explaining why this is true. A further demonstration would be very helpful.

5. I have a further question: Why not consider the line of work on total variation (TV) in non-stationary bandits? There is a rich body of work in that area.

**Questions:**

I would kindly refer to the weakness part. I would appreciate the authors' responses to the main points.

---

> ### Author Response · Authors · 2024-11-23
>
> # Question 1
> If I understand this paper correctly, one of the main contributions of the paper is that the proposed algorithm does not require knowledge of $M$ and achieves a $\sqrt{MT}$ guarantee and $KT$ computation complexity. One of my major concerns is that, based on Table 1, it seems like (Wei & Luo 2021) has studied the setting without knowledge of computation complexity. One of my major concerns is that, based on Table 1, it seems like (Wei & Luo 2021) has studied the setting without knowledge of $M$ , proposing the algorithm Master, which also achieves optimal regret and the same computational complexity. So I am a bit confused by the paper's claim that it studies this novel setting with an optimal guarantee.
>
> ## Ans:
> ### Practical Superiority and Efficiency of Our Approach
> From the results presented in our paper, it is clear that our diminishing exploration (DE) framework stands out as a practical and efficient alternative to Master+UCB. While both approaches achieve the same asymptotic regret performance and computational complexity scaling without requiring prior knowledge of $M$, the key difference lies in their practical usability. Master+UCB relies on a multi-level hierarchical structure to manage unknown changes, maintaining multiple competing instances simultaneously. This design introduces substantial computational overhead and requires long time horizons to be effective. In contrast, our DE framework avoids these complexities, offering a streamlined and computationally efficient solution that performs robustly even with modest time horizons.
> As demonstrated in Figures 1 and 4, our DE approach consistently achieves significantly lower regret and computation time across a variety of scenarios, outperforming Master+UCB even under different values of $M$ and $T$. In particular, Figure 4c showcases how our method excels in runtime efficiency, highlighting its ability to adapt effectively without the need for extensive resources.
> Furthermore, the limitations of Master+UCB in practical applications have also been acknowledged in a recent manuscript (https://arxiv.org/pdf/2410.13772), which emphasizes the need for algorithms like ours that deliver strong performance with an extremely low computational cost. Our DE framework bridges the gap between theoretical guarantees and real-world usability, addressing real-world constraints while maintaining top-tier performance.
> By eliminating unnecessary complexity and focusing on practical adaptability, our approach provides a scalable and computationally efficient solution to real-world challenges. It’s not just about achieving low regret—it’s about delivering real impact where it matters most.

---

> ### Author Response · Authors · 2024-11-23
>
> # Question 2
> Additionally, aside from this single parameter change (exploration rate), it would be helpful to understand what other novelties are introduced. Based on the current modification, it is somewhat unclear how the algorithm stands out in terms of novelty.
>
> ## Ans:
> While many active methods focus primarily on designing or refining the change detection (CD) component, designing an adaptable exploration scheme is equally critical for effectively managing nonstationarity. Our approach introduces a flexible exploration mechanism that can work with various CD algorithms, enhancing adaptability and reducing regret—an important, yet often underexplored, aspect in CD-based methods.
>
> ### A Novel Approach
> As summarized in Table 1, prior to this work, only a handful of active methods, such as GLR-UCB, managed to operate without relying on prior segment knowledge. However, this capability came with a significant trade-off: high computational complexity. Similarly, the MASTER algorithm (Wei & Luo, 2021) avoids the need for segment knowledge, but its practicality diminishes in realistic settings. As noted in recent analyses (https://arxiv.org/abs/2410.13772), MASTER only becomes effective under very large time horizons—a luxury that is often unattainable in practical scenarios.
> This is where our work truly stands out. Unlike MASTER, our algorithm achieves low regret even within modest time horizons, as demonstrated in our simulations. This flexibility makes it far more applicable to real-world problems where abrupt changes occur but data availability or time constraints limit long-term planning. By bridging the divide between theory and application, our approach offers a practical and efficient solution to address real-world constraints while maintaining strong performance.
>
> ### Shift in Focus from Detection to Exploration
> While previous research has concentrated on optimizing CD algorithms, our approach emphasizes a new exploration mechanism that dynamically adapts to changing environments. This shift allows us to achieve a balanced trade-off between exploration and detection delay, improving overall performance to complement CD design.
>
> ### Versatility and Integration with Various CD Algorithms
> By designing the exploration scheme independently of any specific CD algorithm, our framework is highly adaptable and can be combined with multiple CD strategies (e.g., M-UCB, GLR-UCB). This versatility enables our approach to serve as a general enhancement for CD-based methods, supporting a wide range of applications without requiring additional tuning.
>
> In summary, our contribution extends beyond a single parameter change; it lies in designing an adaptable exploration scheme that enhances CD-based methods by making them more versatile and efficient. We hope this clarifies how our work stands out in terms of novelty.
>
> # Question 3
> The second concern is about the choice of $\alpha$. It seems like the paper never explicitly states how to choose the parameter $\alpha$., which, however, is key in the algorithm and used in almost all cases. Also, the regret bound has a dependency on $\alpha$. Therefore, without this, it is hard to quantify how we can deploy this algorithm wit
>
> ## Ans
> The choice of $\alpha$ and its impact on the regret bound is indeed an important aspect of our analysis.
>
> ### Theoretical Guidance on $\alpha$
> We provide theoretical guidance for selecting α\alphaα in Corollaries 4.4 and 4.6 in Appendix C. These results show how $\alpha$ affects the regret bound, and we outline specific choices that ensure optimal performance in terms of $O(\sqrt{KMT \log T})$.
>
> ### Choice of $\alpha$ in Experiments
> For our empirical studies, we simply set $\alpha = 1$ for all experiments, as detailed in Appendix D, without extensive tuning. Our results highlight another important advantage of the proposed algorithm: it does not require excessive fine-tuning of hyperparameters.
>
> By referencing both the theoretical analysis and experimental tuning strategies, we hope to clarify the role of $\alpha$ in our approach and how it can be chosen for both theoretical and practical purposes. We appreciate your feedback on this point and hope this addresses your concern.

---

> ### Author Response · Authors · 2024-11-23
>
> # Question 4:
> The paper lists the statement about the computational complexity in Table 1, which is $KT$. However, it seems like there is no theoretical or formal statement explaining why this is true. A further demonstration would be very helpful.
>
> ## Ans:
> Our algorithm’s complexity can be broken into three components:
> Diminishing Exploration (Lines 3–5): The complexity is $O(T)$.
>
> ### Base Algorithm (Line 10):
> The complexity of this part varies depending on the base algorithm. For UCB, as adopted in M-UCB, UCB values are computed for each of the $K$ arms at each timestep. Over $T$ timesteps, this results in a complexity of $O(KT)$.
>
>
> ### Change Detection (CD) Algorithm:
> The complexity of the CD component again depends on the specific algorithm used. For example, the change detector adopted in M-UCB has a complexity of $O(1)$.
>
> ### Overall Complexity Analysis for M-UCB:
> We provide an example of overall complexity analysis for M-UCB detector with DE as follows:
>
> Complexity analysis of Algorithm 1 (CD-UCB with Diminishing Exploration):
> - The main loop iterates $ T$) times, and within each timestep:
>     - When in DE, the complexity is $O(1)$.
>     - The UCB computation over $ K$ arms has a complexity of $O(K)$.
>     - The change detection step has a complexity of $O(1)$ per timestep.
> - Therefore, the total complexity for Algorithm 1 simplifies to:  $O(T \times K) = O(TK)$
>
> In summary, the computational complexity of M-UCB with DE is $O(KT)$. The above analysis further highlights one of the advantages of the proposed minimalist approach: it can be easily combined with any active method without adding significant computational complexity. Furthermore, the clear separation between DE and the underlying active method allows for an independent complexity analysis.
>
> # Question 5:
> I have a further question: Why not consider the line of work on total variation (TV) in non-stationary bandits? There is a rich body of work in that area.
>
> ## Ans:
> Thank you for pointing out the rich body of work on total variation (TV) in non-stationary bandits. TV is indeed a valuable framework for quantifying changes in the environment, assuming smooth transitions with a bounded total variation over time (Besbes et al., 2014). However, our focus is on scenarios characterized by abrupt, discrete shifts—such as those found in recommendation systems or dynamic pricing—where a piecewise-stationary assumption is more applicable and practical.
>
> These two approaches address different environmental assumptions. TV-based settings are typically well-suited to smoothly changing or adversarial environments, while piecewise-stationary modeling excels in capturing sudden, significant shifts.
>
> In practical domains like recommendation systems (Li et al., 2011, Bouneffouf et al., 2012, Kveton et al., 2014), online advertising (Girgin et al., 2012, Schwartz et al., 2017), and dynamic pricing (Tajik et al., 2024), changes in user behavior or market dynamics often occur abruptly—triggered by trends, external events, or competitive moves. These shifts are rarely gradual; they arrive suddenly and decisively. The piecewise-stationary model captures this reality with precision, making it particularly well-suited for high-impact, fast-evolving environments. By aligning closely with these abrupt transitions, the model provides a robust framework for adapting to the dynamic nature of real-world scenarios.
>
> We believe that our focus on abrupt changes provides a complementary perspective to the broader TV framework, offering tools for scenarios where sudden shifts dominate. Together, these approaches contribute to addressing diverse challenges in non-stationary bandit problems.

---

> ### Author Response · Authors · 2024-12-01
>
> We wanted to kindly follow up on the rebuttal we submitted. We truly appreciate your time and thoughtful feedback, and we wanted to check if you have had the opportunity to review our response.
> We have provided further evidence in another recent post (Gerogiannis et al. 2024) demonstrating the practical challenges associated with MASTER, as highlighted in our work. We would greatly appreciate it if you could let us know if there is anything else that requires clarification or if you have any remaining concerns. Your feedback has been instrumental in shaping our work, and we are truly grateful for your support.

---

### Official Review · Reviewer_rFsn · 2024-11-08

**Soundness:** 3
**Presentation:** 3
**Contribution:** 2
**Rating:** 5
**Confidence:** 4

**Summary:**

This paper revisits a well-studied problem, the piecewise stationary multi-armed bandit problem, and proposes a new exploration scheme, referred to as "diminishing exploration (DE)", which can be used in conjunction with change detection-based approaches. Regret bounds have been developed for the proposed algorithms, and preliminary experiment results are demonstrated in Section 6.

**Strengths:**

- The paper is well-written and the authors clearly present their main ideas.

- The diminishing exploration (DE) approach can be applied to many change detection based algorithms.

**Weaknesses:**

- This paper seems to lack novelty. The standard piecewise stationary bandits have been extensively studied in the past decade. Also, the main idea of this paper, the diminishing exploration (DE), seems to be quite straightforward.

- There is also not much novelty in the analysis techniques. The regret analyses in this paper seem standard.

- The assumptions in this paper also seem restrictive. For instance, Assumption 4.2 seems both unnatural and unnecessary. It seems that some assumptions (e.g. Assumption 4.2) are from previous literatures, but it is better to get rid of such unnatural and unnecessary assumptions. Can the authors develop regret bounds for DE without such restrictive assumptions? Please explain.

**Questions:**

Please try to address the weaknesses above.

---

> ### Author Response · Authors · 2024-11-23
>
> # Quesiton 1
> This paper seems to lack novelty. The standard piecewise stationary bandits have been extensively studied in the past decade. Also, the main idea of this paper, the diminishing exploration (DE), seems to be quite straightforward.
>
> ## Ans:
> We understand that the idea of diminishing exploration (DE) may appear straightforward at first glance, but we would like to emphasize several aspects of our approach that contribute to its novelty and practical significance.
>
> (i) A Novel Approach: As summarized in Table 1, prior to this work, only a handful of active methods, such as GLR-UCB, managed to operate without relying on prior segment knowledge. However, this capability came with a significant trade-off: high computational complexity. Similarly, the MASTER algorithm (Wei & Luo, 2021) avoids the need for segment knowledge, but its practicality diminishes in realistic settings. As noted in recent analyses (https://arxiv.org/abs/2410.13772), MASTER only becomes effective under very large time horizons—a luxury that is often unattainable in practical scenarios.
> This is where our work truly stands out. Unlike MASTER, our algorithm achieves low regret even within modest time horizons, as demonstrated in our simulations. This flexibility makes it far more applicable to real-world problems where abrupt changes occur but data availability or time constraints limit long-term planning. By bridging the divide between theory and application, our approach offers a practical and efficient solution to address real-world constraints while maintaining strong performance.
>
>
> (ii) Shift in Focus from Detection to Exploration: While much of the existing work on piecewise-stationary bandits has focused on optimizing change detection (CD) mechanisms, we shift the focus to the design of an adaptable exploration strategy. This is not a trivial change—it represents a critical rethinking of how to balance exploration and exploitation in nonstationary environments. By dynamically adjusting exploration rates, we complement CD methods and enhance overall performance, enabling our algorithm to adapt effectively to a variety of scenarios.
>
> (iii) Dynamic Decreasing Exploration (Instead of Increasing Exploration) Within a Segment: One of the most exciting aspects of our approach is the concept of dynamically decreasing exploration within a segment. While at first glance it might seem straightforward, the choice between dynamically increasing or decreasing exploration within a segment is far from trivial. In fact, as noted by another reviewer, increasing exploration over time could make sense in environments where the likelihood of a new change grows with time. But here’s the catch: in long segments, an increasing exploration rate quickly becomes excessively large, causing significant exploration costs. We extensively analyzed the trade-offs involved, and our choice of a diminishing approach was carefully calculated to ensure it balances exploration cost with regret minimization. Moreover, determining the precise rate at which exploration should diminish required detailed theoretical analysis to achieve optimal performance.
>
> (iv) Versatility and Integration with Various CD Algorithms: By designing the exploration scheme independently of any specific CD algorithm, our framework is highly adaptable and can be integrated with multiple CD strategies (e.g., M-UCB and GLR-UCB). This versatility enables our approach to serve as a general enhancement for CD-based methods, supporting a wide range of applications without requiring additional tuning.
>
> In summary, while DE may seem straightforward in concept, its design involves critical decisions regarding the dynamics of exploration adjustment and the rate of decrease, which are highly non-trivial and require precise theoretical and practical analysis. Our contribution extends beyond a single parameter change; it lies in designing an adaptable exploration scheme that enhances CD-based methods, making them more versatile and efficient. We hope this clarifies how our work stands out in terms of novelty.

---

> ### Author Response · Authors · 2024-11-23
>
> # Question 2:
> There is also not much novelty in the analysis techniques. The regret analyses in this paper seem standard.
>
>
> ## Ans:
> We want to highlight two key innovations in our approach that contribute to the novelty of our work:
>
> (i) Incorporation of Diminishing Exploration in Analysis: A significant aspect of our analysis involves deriving the relationship between the number of samples collected due to diminishing exploration and the time step. We formalized this relationship into a lemma (see Lemma C.2), which provides a theoretical foundation for understanding how diminishing exploration balances exploration and exploitation over time. This is a novel addition that enables us to extend the regret analysis to algorithms using dynamically adjusted exploration rates, filling a gap in existing frameworks.
>
> (ii) Extension via the Skipping Mechanism: In addition to the core framework, we introduced a skipping mechanism that extends the original architecture. This mechanism identifies and skips certain situations where a reset may not be necessary, reducing unnecessary resets and improving the efficiency of the algorithm. Our analysis incorporates this extension, showing how it affects the regret bounds and computational complexity.
>
> By combining these contributions, we believe our analysis goes beyond the standard techniques and offers novel insights into regret analysis for algorithms using diminishing exploration and enhanced change detection mechanisms. We hope this addresses your concern and highlights the novelty in our theoretical framework.
>
> # Question 3:
> The assumptions in this paper also seem restrictive. For instance, Assumption 4.2 seems both unnatural and unnecessary. It seems that some assumptions (e.g. Assumption 4.2) are from previous literatures, but it is better to get rid of such unnatural and unnecessary assumptions. Can the authors develop regret bounds for DE without such restrictive assumptions? Please explain.
>
> ## Ans:
> - Context for Assumption 4.2: Assumption 4.2, which assumes a lower bound on the minimum detectable change $\delta$, is indeed inherited from previous literature (e.g., [Cao et al., 2019]) to ensure the reliable performance of the change detection mechanism. This assumption allows us to select appropriate values for parameters like the window size $w$ and threshold $b$ in the change detection process, thereby guaranteeing the effectiveness of our regret bounds under abrupt changes. Without this assumption, there would be a risk that small changes might go undetected, potentially leading to unbounded regret in theory.
>
> - Empirical Robustness: It is also worth mentioning that, in practice, the diminishing exploration (DE) framework has demonstrated robustness even when Assumption 4.2 is violated. For example, as shown in our simulations (Figure 3(a) in Section 6), the algorithm performs well in environments where the actual minimum change $\delta$ is smaller than the assumed lower bound. These results indicate that the assumption is primarily necessary for theoretical proofs and regret guarantees, while the practical applicability of our algorithm is not significantly impacted by minor violations of this assumption. In other words, in practice, only a rough estimation of $\delta$ suffices.

---

> > ### Comment · Reviewer_rFsn · 2024-12-03
> > **Thanks!**
> >
> > I would like to thank the authors for the detailed responses. I think the responses have partially addressed my concerns.
> >
> > However, I still feel that Assumption 4.2 and 4.3 are very restrictive and unnatural. Assumption 4.2 is made for a particular type of analysis to go through, and as the authors' new experiment results have indicated, even without this assumption, the algorithm still works fine. This suggests that the current analysis is limited, and the authors should provide a new version of analysis without this assumption.
> >
> > Also, Assumption 4.3 requires each segment length is long enough. Intuitively, this is also not necessary. If a segment length is short, even if the agent fails to detect this segment, the incurred regret is still very small, right?
> >
> > Due to the above concerns, and after reading other reviews, I prefer to keep my score.

---

> ### Author Response · Authors · 2024-12-01
>
> We wanted to kindly follow up on the rebuttal we submitted. We truly appreciate your time and thoughtful feedback, and we wanted to check if you have had the opportunity to review our response.
> We have specifically addressed your concerns regarding the novelty of our work. We would like to kindly ask if there are any remaining aspects that need further clarification or if you have any additional concerns we should address. Your input is invaluable, and we deeply appreciate your guidance in improving our work.

---

> ### Author Response · Authors · 2024-12-04
>
> We thank the reviewer for the justification. However, we respectfully disagree with it. Our current analysis indeed relies on the two assumptions mentioned by the reviewer. But, none of the existing algorithms is exempt from assumptions, including MASTER. We fully agree with the reviewer that achieving optimal results while completely removing such assumptions would be a significant breakthrough. Nevertheless, we believe that these assumptions should not overshadow the significant contributions of our scheme, which include:
> * Provable optimality without requiring knowledge of $𝑀$;
> * Numerical dominance over {\it all} existing schemes (including MASTER);
> * Compatibility with {\it any} active method; and
> * Easy extensibility to address non-optimal-arm changes.
>
> Furthermore, our numerical results demonstrate that even when the two assumptions are violated, the proposed algorithm still delivers excellent performance. This suggests the potential for eventually relaxing these assumptions. Rather than seeing this as a critique of the effectiveness of our proof, we believe it should be viewed as an affirmation that our algorithm performs robustly even without such assumptions, further underscoring the contribution of our work. Given that a recent article by Gerogiannis et al. (2024) highlights the insufficiency of MASTER, we firmly believe that our work is both timely and addresses a critical issue in the piecewise stationary bandit problem.

---

### Official Review · Reviewer_mdQN · 2024-11-10

**Soundness:** 3
**Presentation:** 3
**Contribution:** 3
**Rating:** 5
**Confidence:** 3

**Summary:**

This paper considers stochastic multi-armed bandits with structured nonstationarity, specifically where the arms change value at $M$ points throughout the time horizon $T$ (where $M$ is unknown) but are otherwise stationary. The authors develop a new algorithm to handle the exploration aspect of this problem, which they combine with an existing change-detection algorithm.

They show theoretically that this approach achieves sublinear, $\mathcal{O}(\sqrt{MT})$ regret with computational complexity $\mathcal{O}(KT)$, and show strong empirical performance on a simulator based on a public dataset from Yahoo.

**Strengths:**

1. Nonstationarity is an important challenge for multi-armed bandits essential to making this approach useful for a broad range of real-world problems. This paper is part of the literature on "active methods" in piecewise-stationary bandits that actively try to detect and adapt to change points. This paper overcomes a limiting assumption in existing work, which is the assumption that the number of change points are known a priori (thus the existing algorithms could tune specifically with that knowledge). This advance could help make MAB solvers with piecewise stationarity more useful.

2. The authors propose a novel approach for exploration, using a "diminishing exploration" approach where the amount of exploration decays with time. This method does not require prior knowledge of the number of change points $M$, and can be paired with a range of existing change detection algorithms. The algorithm moves from an exploration stage, then update initial estimates of $u$, then act following a UCB approach (based on M-UCB), while doing a per-timestep check for change detection. The authors add a heuristic that adds a skipping mechanism (algorithm 2) to ignore potentially unnecessary arms, with a tuning parameter $\eta$ that describes how conservative to be in skipping.

3. The authors show theoretically that the regret is bounded by $\tilde{\mathcal{O}} \sqrt{MKT})$. This result is found by studying the number of times the optimal arm changes $S$ (referred to as the number of "super-segments").

**Weaknesses:**

1. The choice to model nonstationarity as a series of piecewise-stationary with a fixed number of change points is not very well justified. What are some examples for why this assumption would arise, rather than a smoother type of nonstationarity?

2. Related to #1, this paper compares to other literature in piecewise-stationary bandits (theoretically and empirically), but there is a greater amount of literature on nonstationary bandits. One possible analysis that could help strengthen the usefulness of this paper is to empirically evaluate your method on more general nonstationary settings, and compare your performance with existing methods for nonstationary bandits. I imagine this would be a more realistic model for the vast array of real-world settings. If you are able to perform well even in those settings, that would be extremely useful. However even if not, it would still be useful to show/quantify.

3. Algorithm 2 (the skipping mechanism) is a heuristic approach, but the importance of this heuristic is not studied. It would help to include an ablation to shows the impact of this heuristic --- how much of the improved performance depends on this?

4. GLR-UCB seems to perform very similarly with and without your diminishing exploration approach? Same as CUSUM-UCB, with and without diminishing exploration, as T scales and on the Yahoo dataset.


*Detailed comments*

There are small grammatical things that could be improved throughout the paper.
- line 21: "despite oblivious" -> "despite being oblivious"
- line 23: delete "the"
- line 36: delete both "the"s


Small issues:
- line 213: "however, in the other sense." ends the sentence. This is an incomplete sentence?
- line 220: "a perfect balance" -> "an optimal balance"

- in Intro, M-UCB and GLR-UCB is not defined or described.
- Plots are very small and hard to read, especially to distinguish the line and clarify which approaches are your vs. existing methods.

**Questions:**

1. Experiments describe that evaluations beyond 20,000 timesteps was infeasible because of long runtime. But Figure 4d appears to show up to $T=200,000$?

2. The M-UCB and GLR-UCB approaches that this algorithm builds off of does not seem to be described? How do they work? What does the "M" in M-UCB stand for?

---

> ### Author Response · Authors · 2024-11-23
>
> # Quension 1
> The choice to model nonstationarity as a series of piecewise-stationary with a fixed number of change points is not very well justified. What are some examples for why this assumption would arise, rather than a smoother type of nonstationarity?
>
> ## Ans:
> We believe that the abrupt-change, piecewise-stationary framework is well-suited for many practical applications and serves as a flexible model that can encompass various types of nonstationarity:
>
> Nonstationarity Comes in Two Flavors: Nonstationary environments can typically be categorized into two distinct branches—slowly changing and abruptly changing. These are fundamentally different worlds: abrupt changes are sharp, clear-cut shifts, while slow changes are smooth and continuous. The piecewise-stationary model dives deep into the world of abrupt changes, where transitions happen as discrete jumps. This framework isn’t just practical—it’s powerful. By explicitly focusing on these discrete shifts, it allows us to build algorithms that excel in environments with rapid, sudden changes.
>
> Application-Specific Suitability: In practical domains like recommendation systems (Li et al., 2011, Bouneffouf et al., 2012, Kveton et al., 2014), online advertising (Girgin et al., 2012, Schwartz et al., 2017), and dynamic pricing (Tajik et al., 2024), changes in user behavior or market dynamics often occur abruptly—triggered by trends, external events, or competitive moves. These shifts are rarely gradual; they arrive suddenly and decisively. The piecewise-stationary model captures this reality with precision, making it particularly well-suited for high-impact, fast-evolving environments. By aligning closely with these abrupt transitions, the model provides a robust framework for adapting to the dynamic nature of real-world scenarios.
>
> Piecewise-Stationary as a General Framework: Piecewise-stationary modeling can be viewed as a general case for handling nonstationarity:
> - If $M=1$ (only one segment), the model reduces to the standard stochastic bandit problem.
> - If $M=T$ (changes at every timestep), the model aligns with adversarial bandits.
>
> In summary, we see piecewise-stationary modeling not only as a practical choice for handling abrupt changes but also as a general framework encompassing a wide range of nonstationary environments. We hope this clarifies our modeling choice and its relevance to various scenarios.
>
>
> # Quension 2
> Related to #1, this paper compares to other literature in piecewise-stationary bandits (theoretically and empirically), but there is a greater amount of literature on nonstationary bandits. One possible analysis that could help strengthen the usefulness of this paper is to empirically evaluate your method on more general nonstationary settings, and compare your performance with existing methods for nonstationary bandits. I imagine this would be a more realistic model for the vast array of real-world settings. If you are able to perform well even in those settings, that would be extremely useful. However even if not, it would still be useful to show/quantify.
>
> ## Ans:
> To explore the applicability of our approach in more general nonstationary settings, we further conducted experiments in a slowly changing environment, where the two arms followed sinusoidal oscillations as considered in (Qi et al., 2023). This setup represents a gradual, continuous shift, distinct from the abrupt changes in our primary focus.
>
> Performance of Active and Passive Methods: Surprisingly, some active methods perform reasonably well in this setting. However, their effectiveness is highly dependent on the choice of change detector. For example, M-UCB struggles in this environment, as it is not designed for smooth transitions. Among the passive methods (e.g., Discounted-klUCB and Discounted-klUCB-TS), setting a discount factor of 0.99 yields strong results, emphasizing their adaptability to slowly changing environments.
>
> Challenges with Constant Exploration Rates: A notable limitation of standard active methods is their dependency on $M$ (the number of changes) to define an appropriate constant exploration rate. This dependency creates ambiguity in environments with gradual transitions, as $M$ becomes difficult to define. By contrast, our diminishing exploration (DE) approach is independent of $M$, eliminating the need for such assumptions and enabling a more robust and flexible adaptation.
>
> Performance of Our Diminishing Exploration Approach: Despite the challenges of a slowly changing environment, our DE algorithm demonstrates slightly better performance than most standard active methods, except M-UCB. This highlights the adaptability of our method even in settings outside its primary focus of abrupt changes.
>
> Figure of slowly change environment: https://imgur.com/a/slow-change-environment-Q76TBxB
> Figure of slowly change environment regret: https://imgur.com/a/slowly-change-regret-b9UPCsV

---

> ### Author Response · Authors · 2024-11-23
>
> # Question 3:
> Algorithm 2 (the skipping mechanism) is a heuristic approach, but the importance of this heuristic is not studied. It would help to include an ablation to show the impact of this heuristic --- how much of the improved performance depends on this?
>
> ## Ans:
> Theoretical Justification: The skipping mechanism is not purely heuristic; it has a theoretical foundation, as shown in our proof of regret bounds (see Theorem C.13 in Section C of the Appendix). We want to apologize for a typo in the original manuscript, where this equation was mistakenly referred to as Theorem 4.1. Specifically, the skipping mechanism ensures that the exploration phase is effectively balanced with the need to avoid unnecessary resets caused by minor fluctuations in rewards, thereby improving both computational efficiency and regret minimization.
>
> Empirical Evidence: Our empirical results further support the importance of the skipping mechanism. As illustrated in the following figure, the inclusion of the skipping mechanism significantly improves the performance of the algorithm in regret. By applying the proposed diminishing exploration and the skipping mechanism, we can improve the empirical regrets of various active methods, including M-UCB, CUSUM-UCB, and GLR-UCB.
>
> Regret with Skipping Mechanism: https://imgur.com/a/Sqr845R
>
> # Question 4
> GLR-UCB seems to perform very similarly with and without your diminishing exploration approach? Same as CUSUM-UCB, with and without diminishing exploration, as T scales and on the Yahoo dataset.
>
> ## Ans:
> The performance of GLR-UCB and CUSUM-UCB with our diminishing exploration (DE) mechanism appears similar to their standard versions, especially as $T$ scales and on the Yahoo dataset. However, the proposed DE mechanism can indeed improve GLR-UCB and CUSUM-UCB in various scenarios and hence provides meaningful advantages, particularly in terms of flexibility and adaptability. Specifically:
>
> (i) GLR-UCB and CUSUM-UCB On $M$-Scaling: As shown in Figure 3(b) in our paper, the DE mechanism can improve both GLR-UCB and CUSUM-UCB for a better scaling in the number of segments $M$.
>
> (ii) Flexibility Without Knowledge of $M$: One of the primary advantages of the DE mechanism is that it allows CUSUM-UCB to function effectively without needing prior knowledge of $M$ (the number of change points), which is typically required to tune parameters optimally. This makes CUSUM-UCB more flexible and applicable to a wider range of scenarios where $M$ is unknown or difficult to estimate. Even for those scenarios where the performance in terms of regret is similar, the ability to remove dependency on $M$ is a practical improvement, as it simplifies the setup and reduces the need for parameter tuning.
>
> (iii) GLR-UCB On $T$-Scaling: While the differences in performance may appear small at first glance, our results demonstrate statistically significant improvements. Specifically, the gap between the regret of the methods with and without DE exceeds the combined standard deviations of the individual methods, confirming that the improvement is not due to random variance.
> A zoomed-in view of Figure 3(c) is available at the following link https://imgur.com/a/YDdYMAf.
> This indicates that the DE mechanism consistently enhances performance as $T$ scales, even if the differences appear subtle visually.
>
> (iv) GLR-UCB On the Yahoo Dataset: While the performance differences on the Yahoo dataset appear minimal, this is likely due to the dataset's specific characteristics. To further explore our approach, we retain the mean reward structure but introduce controlled variations with an even larger horizon $T=54000$. These experiments reveal a clear advantage of our DE framework, particularly in more dynamic scenarios. Here is the figure link https://imgur.com/a/vIpgAqZ.
>
> Regret of Yahoo Dataset with a larger horizon $T$: https://imgur.com/a/vIpgAqZ
>
> This demonstrates DE’s ability to adapt and excel in environments with varying change dynamics, highlighting its versatility and practical relevance for real-world applications like recommendation systems and dynamic pricing.
>
> # Question 5
> Experiments describe that evaluations beyond 20,000 timesteps was infeasible because of long runtime. But Figure 4d appears to show up to $T=200000$?
>
> ## Ans:
> To clarify: when we stated that "evaluations beyond 20,000 timesteps were infeasible due to long runtime" (lines 527–528), we were specifically referring to the AdSwitch, ArmSwitch, and Meta algorithms, not the MASTER+UCB and M-UCB (DE) algorithms shown in Figure 4(d). Figure 4(d) presents the relative execution time ratio of MASTER+UCB to M-UCB (DE) at larger $𝑇$ values, extending up to $T=200000$.

---

> ### Author Response · Authors · 2024-11-23
>
> # Question 6:
> The M-UCB and GLR-UCB approaches that this algorithm builds off of does not seem to be described? How do they work? What does the "M" in M-UCB stand for?
>
> ## Ans:
> Brief Overview of M-UCB and GLR-UCB: Both M-UCB and GLR-UCB are change-detection-based approaches developed for nonstationary bandit problems. M-UCB (Monitor-UCB) incorporates a "monitor" mechanism for change detection, which segments the reward data into blocks and compares the empirical means across blocks to detect abrupt shifts in reward distributions. When a significant change is detected, the UCB estimates are reset.
>
> In contrast, GLR-UCB uses the Generalized Likelihood Ratio (GLR) test as its detection mechanism. The GLR statistic identifies potential change points by assessing deviations in observed rewards, and, upon detection, triggers a reset in UCB estimates to adapt to the new reward distribution.
>
> Meaning of "M" in M-UCB: The "M" in M-UCB stands for "Monitor," highlighting the algorithm's focus on actively monitoring reward changes. This method uses change detection to adaptively partition the reward sequence into stationary segments, adjusting to changes in the environment.

---

> ### Comment · Reviewer_mdQN · 2024-11-26
>
> Thank you for your thoughtful responses. The justification of piece-wise linear seems more clear now. I think your response this would be valuable to add to your manuscript.
>
> Thank you as well for your additional analysis. However, none of the imgur links work; they all take me to the imgur homepage.

---

> ### Author Response · Authors · 2024-11-26
>
> We sincerely apologize for the issues with the figure links in our initial submission. We have corrected the links and provided the updated figures below:
>
> The figure of the slowly changing environment:
> https://imgur.com/a/slow-change-environment-Q76TBxB
>
> The figure of slowly changing environment regret:
> https://imgur.com/a/slowly-change-regret-b9UPCsV
>
> Regret with Skipping Mechanism:
> https://imgur.com/a/Sqr845R
>
> Regret of Yahoo Dataset with a larger horizon $T$:
> https://imgur.com/a/vIpgAqZ
>
> A zoomed-in view of Figure 3(c):
> https://imgur.com/a/YDdYMAf
>
> We sincerely hope these updated figures satisfactorily address your questions/comments above.

---

> ### Author Response · Authors · 2024-12-01
>
> We wanted to kindly follow up on the rebuttal we submitted. We truly appreciate your time and thoughtful feedback, and we wanted to check if you have had the opportunity to review our response.
> We are glad to inform you that the figures are now accessible via the links provided. Once again, we sincerely thank you for pointing out this issue. If you have any further comments or if there’s anything else you’d like us to clarify, please don’t hesitate to let us know.

---

### Public Comment · ~Argyrios_Gerogiannis1 · 2024-11-25
**Reference Suggestion**

Dear Authors of Submission 5906,

Thank you for this interesting work on Piecewise Stationary Multi-Armed Bandits and for discussing our work on MASTER (https://arxiv.org/abs/2410.13772) in the reviewer discussions. We appreciate your acknowledgment of its relevance to your study.

To further help readers understand MASTER's limitations, I kindly suggest citing our paper, "Is Prior-Free Black-Box Non-Stationary Reinforcement Learning Feasible?" in the next version of your paper. I believe it would complement your research and provide additional context.

Thank you for considering this suggestion.

---

> ### Author Response · Authors · 2024-11-25
> **Acknowledgment and Citation Update Confirmation**
>
> Thank you for your message. We appreciate your suggestion and agree that citing "Is Prior-Free Black-Box Non-Stationary Reinforcement Learning Feasible?" would add valuable context and complement my research. We will make sure to include it in the next version of the paper.
>
> Thank you again for your thoughtful recommendation!

---

> ### Author Response · Authors · 2024-11-29
> **Reference added**
>
> Thanks again for your suggestion. We have uploaded a new version of our paper, which cites the recommended manuscript in page 2 (line 088).

---

### Author Response · Authors · 2024-11-26

Thanks to the reminder by Reviewer mdQN, we became aware that there were typos in our previous links to the new figures. We have corrected the links and provided the updated figures below:

The figure of the slowly changing environment:
https://imgur.com/a/slow-change-environment-Q76TBxB

The figure of slowly changing environment regret:
https://imgur.com/a/slowly-change-regret-b9UPCsV

Regret with Skipping Mechanism:
https://imgur.com/a/Sqr845R

Regret of Yahoo Dataset with a larger horizon $T$:
https://imgur.com/a/vIpgAqZ

A zoomed-in view of Figure 3(c):
https://imgur.com/a/YDdYMAf

---

### Author Response · Authors · 2024-11-28

We sincerely thank all the reviewers for their valuable feedback, which has greatly contributed to improving our work. In response to your comments, we have revised the paper, and we believe the latest version represents a significant enhancement. Below, we summarize the key improvements:

- We corrected typos and grammatical issues throughout the paper to improve clarity and readability.

-  The introduction has been refined with more detailed descriptions and additional references, resulting in a smoother and more comprehensive presentation.

- We completed the parameter configurations for the experiments and added a table to enhance readability and accessibility.

* Due to space constraints, we were unable to enlarge the figures in Section 6. However, we have included zoomed-in versions of the simulation figures in Appendix G for better visualization.

* M-UCB and GLR-UCB algorithms are now explicitly explained in the Introduction.

* A new reference (Gerogiannis et al. 2024) has been added to highlight that MASTER may not perform well in practical scenarios, providing stronger motivation for our proposed approach.

- All new experimental results on piecewise stationary scenarios generated during the rebuttal period have been included in Appendix F.

- The extension section has been slightly revised to clarify that the underlying change detection (CD) algorithm must be reset each time a (not necessarily optimal-arm) change is detected.

We deeply appreciate your thoughtful feedback, which has been invaluable in refining and improving our paper. We sincerely hope the reviewers can see the merits in devising a minimalist approach and rigorously proving that this scheme can perform near optimally with extremely low complexity. Thank you for your time and effort.

---

### Meta-Review · Area_Chair_hfV4 · 2024-12-20

**Metareview:**

There are various concerns raised about the novelty and relevance of the setting. In my opinion these concerns are of minor relevance. However, the more serious issue is the comparison to MASTER. A lack of novelty is not that important if there are clear cut advantages to new methods. However, in this paper the improvement shows only in experiments and not in the regret or complexity analyses. I believe that more is needed and that the paper needs to improve on that front to make it publishable in a top conference.

**Additional Comments On Reviewer Discussion:**

The key point about the comparison to MASTER were discussed but it is not possible to argue against that shortcoming. The paper needs a major revision to alleviate these concerns.

---

### Decision · Program_Chairs · 2025-01-22

Reject